# Multiverse: Your Language Models Secretly Decide How to Parallelize and Merge Generation

**Xinyu Yang**[*†], **Yuwei An**[*†], **Hongyi Liu**[†], **Tianqi Chen**[†‡], **Beidi Chen**[†]

[†]CMU, [‡]Nvidia

xinyuya2, yuweia, hongyil2, tqchen, beidic@andrew.cmu.edu

https://Multiverse4FM.github.io

## Abstract

Autoregressive Large Language Models (AR-LLMs) frequently exhibit implicit parallelism in sequential generation. Inspired by this, we introduce **Multiverse**, a new generative model enabling natively parallel generation. Multiverse internalizes a MapReduce paradigm, generating automatically through three stages: (i) a Map stage for adaptive task decomposition, (ii) a Process stage for parallel subtask execution, and (iii) a Reduce stage for lossless result synthesis. Next, we build a real-world Multiverse reasoning model with co-design of data, algorithm, and system, enabling rapid and seamless transfer from frontier AR-LLMs. For data creation, we develop *Multiverse Curator*, an automated LLM-assisted pipeline that transforms sequential reasoning chains into structured training data, avoiding costly human annotations. Algorithmically, we design *Multiverse Attention* to separate parallel reasoning steps while keeping compatibility with causal attention for efficient training. Systematically, we implement *Multiverse Engine* to support parallel inference. It features a dedicated interpreter that dynamically switches between sequential and parallel generation, triggered directly by the model. After a 3-hour fine-tuning with 1K examples, *our Multiverse-32B stands as the only open-sourced non-AR model achieving performance on par with leading AR-LLMs of the same scale*, evidenced by AIME24 & 25 scores of 54% and 46%, respectively. Moreover, our budget control experiments show that Multiverse-32B exhibits superior scaling, outperforming AR-LLMs by 1.87% on average using the same context length. Such scaling further leads to practical efficiency gain, achieving up to $2\times$ speedup across varying batch sizes. We have open-sourced the entire Multiverse ecosystem, including data, model weights, serving system, supporting tools, as well as data curation prompts and detailed training and evaluation recipes.

## 1 Introduction

> *"In an infinite multiverse, everything that can happen does happen—somewhere."*

Test-time scaling has advanced Large Language Models (LLMs) by increasing the generation length [20, 16] and depth [12], closely reflecting human cognition. However, empowered by modern hardware like GPUs, ideal LLMs can surpass humans by scaling a third dimension: *width*, which allows parallel task-solving. Realizing this potential requires LLMs to "smartly" parallelize and merge their generation, following the classic *MapReduce* paradigm [8]: splitting into subtasks, processing them independently in parallel, and merging their results. Such philosophy has a long history in computer science [26, 1] while driving fundamental progress in other fields including manufacturing [19], agriculture [28], and finance [7]. This shift from sequential to parallel task-solving unlocks economies of scale: reducing the time per unit and keeping near-constant overall latency as task complexity grows, thereby offering a promising path towards artificial superintelligence (ASI).

39th Conference on Neural Information Processing Systems (NeurIPS 2025).

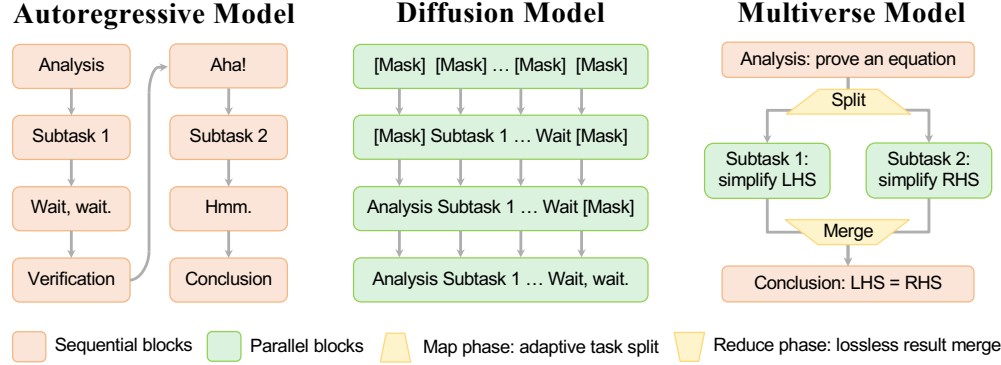

Figure 1: **Model Overview.** Autoregressive models are constrained to sequential generation, while diffusion models struggle with coherence in parallel generation. Our **Multiverse** is a new generative model natively built on the MapReduce paradigm, enabling adaptive and lossless parallel generation.

Despite this potential, current LLMs are limited by the inherently sequential nature of autoregressive (AR) generation. While non-AR architectures, such as diffusion models [37, 49] and consistency models [22], along with their hybrid semi-AR variants [2, 29], natively enable parallel generation, they incur substantial computational waste. Their rigid, brute-force parallelism ignores logical dependencies, partly due to a lack of real-world training data to supervise when and how parallel generation should occur. Another stream of research [48, 6, 45, 32] leverages external tools to parallelize or merge tasks heuristically, leading to the loss of internal states, like the intermediate reasoning steps, during communication with external modules. Although our concurrent work [21, 36] allows internal communication, they introduce inconsistencies between training and inference, limiting their effectiveness to short sequences with shallow parallelism. These challenges raise a question: *How to design a modeling framework for LLMs that can (i) adaptively split and merge tasks, (ii) losslessly preserve internal states, and (iii) generally apply to diverse parallelism patterns?*

Due to the dominance of AR-LLMs, we start to answer it by revealing *numerous intrinsic parallelism in their sequential outputs*. Specifically, we analyze the long Chain-of-Thought (CoT) trajectories from the s1K-1.1 dataset [27]. Among them, over 98% exhibit parallelizable branches, despite being trained only for sequential generation. These branches, as shown in Figure 2, fall into collective and selective ones that appear frequently within individual CoT trajectories, either consecutively or recursively, covering a wide range of scenarios. However, our prompting and probing tests verify that AR-LLMs cannot actively enforce or discern such parallelism. These findings motivate the design of a new modeling framework that can be bootstrapped directly from pre-trained AR-LLMs, which further requires us to address three practical limitations: (i) *Data*: Real-world CoT trajectories lack explicit parallel structure. (ii) *Algorithm*: Transformers with causal attention are limited to sequential generation. (iii) *System*: Inference engines for AR-LLMs cannot support practical parallel generation.

To achieve these, we introduce **Multiverse**, a generative modeling framework built on the MapReduce paradigm that dynamically adjusts its parallelism during generation. It internalizes a three-stage pipeline: a sequential Map stage performs adaptive task decomposition; a parallel Process stage allows independent subtask execution; and a sequential Reduce stage ensures lossless result synthesis. Moreover, the pipeline can invoke itself recursively, enabling optimal time complexity with unlimited resources. We theoretically prove this optimality on a synthetic NP-hard SAT problem, demonstrating that Multiverse is the only framework that achieves a linear-time solution. Based on this, we co-design our data, algorithm, and system, providing a general solution to building a real-world Multiverse model for complex reasoning tasks, offering a smooth and rapid transition from pre-trained AR-LLMs.

**Data Curation.** In Section 5.1, we develop **Multiverse Curator**, an automated LLM-assisted pipeline that transforms sequential reasoning chains into parallel structures via five steps: (i) parsing the sequential chain into a summary tree; (ii) identifying parallelizable nodes within the summary tree; (iii) reformatting the summary into a parallel generation structure; (iv) refilling original reasoning steps into this structure; and (v) adding Map & Reduce stages while rewriting Process stage. Moreover, content and grammar checks are performed to flag low-quality data for regeneration, avoiding costly

manual filtration and annotation. In practice, this process results in **Multiverse-1K**, a dataset of 1,000 high-quality structured training samples for advancing LLM reasoning.

**Algorithm Design.** In Section 5.2, we design **Multiverse Attention** to enable parallel generation while maintaining training efficiency. This is achieved by modifying attention masks and position indices to strictly separate independent reasoning branches in attention calculation. Due to its which can be trained in parallel, similar to causal attention. This design also excels in data efficiency: since these changes are minor, pre-trained AR models can be rapidly transferred from causal attention to Multiverse attention using only a few thousand examples.

**System Implementation.** In Section 5.3, we implement **Multiverse Engine** featuring a specialized interpreter to support MapReduce in execution. By interpreting control tags generated by the Multiverse model itself, our engine can dynamically switch between sequential and parallel generation with near-zero overhead, yielding a flexible workflow. This process includes two stages: (i) Sequential → Parallel: mapping subtasks to separate branches for parallel execution with prefix sharing, and (ii) Parallel → Sequential: reducing Key-Value (KV) states from all branches back into one sequence.

The integration of these modules enables highly efficient training and inference of Multiverse models. Specifically, we develop Multiverse-32B by supervised fine-tuning (SFT) Qwen-2.5-32B-Instruct with only 1K examples, which takes only 3 hours. Empirically, Multiverse-32B achieves significant performance improvement, outperforming the base model by 23.6%, with AIME24 and AIME25 scores of 53.8% and 45.8%, respectively. These results are comparable to AR-LLMs, confirming that Multiverse does not compromise model performance. Furthermore, Multiverse-32B exhibits more efficient test-time scaling, yielding an average improvement of 1.87% within fixed latency constraints. This efficiency stems from its parallel generation capabilities, leading to up to $2\times$ wall-clock speedup per generated token while keeping effective scaling across variable batch sizes range from 1 to 128.

# 2 Related Work

**Test-time Scaling.** Prior work has shown that optimizing AR-LLMs to generate longer outputs improves their reasoning abilities. This is evident in frontier reasoning models built with reinforcement learning (RL) [31, 9, 30, 15, 42], and also validated through supervised fine-tuning (SFT) on smaller models with a few distilled examples [27, 47]. However, this length scaling greatly increases latency due to the sequential nature of AR generation. Other methods like depth scaling [12, 49] suffer from the same issue, while width scaling [5, 32] requires external information to split or merge generations.

**Internal Parallel Generation.** Recent work has increasingly explored other models to replace the commonly used AR models, thereby enabling parallel generation. Among them, discrete diffusion models [37, 38, 3, 23, 40], including masked and absorbed variants, are gaining growing attention. To narrow their gap with AR models, efforts have been made on methods like hybrid AR-diffusion generation [2, 10] and training/test-time scaling [29, 49, 46]. However, [11] has theoretically shown that these approaches cannot reduce the number of sequential generating or sampling steps, as they brute-force parallelize token generation without adhering to inherent relations. Similarly, other work explores continuous diffusion models [4] and consistency models [22]. Among these open-sourced, non-AR models, a common issue is their current inability to scale to complex reasoning tasks, such as AIME [24]. While our concurrent work [21, 36] begins to explore the use of customized attention masks for parallel generation, their design are not general or adaptive, limiting their effectiveness to shallow, non-nested parallelism. In contrast, Multiverse offers a more efficient and scalable approach to enable internal parallel generation, which is generally applicable to diverse parallelism patterns.

**External Parallel Generation.** In another line of research, several approaches leverage external tools or models to enable parallel generation [45, 32, 41, 5, 48]. However, these methods generally leverage heuristic rules and external tools to parallelize or merge their generation. For instance, Best-of-N [5] and self-consistency [41] use a brute-force approach by parallelizing generation at the beginning of generation. Other methods like Monte Carlo tree search (MCTS) [48] and Tree of Thoughts (ToT) [45] offer more fine-grained parallelism, yet they are still fundamentally guided by heuristics and depend on an external verifier. While recent work [32] enables more adaptive parallel generation, it suffers from significant information loss when parallelizing and merging its generation, as it requires inter-model communication when switching between sequential and parallel generation, during which short text summaries rather than complete KV states can be shared between models.

# 3 Long CoT Generation: Sequential or Parallel in Logic?

In this section, we present several key observations of intrinsic parallelism within AR-LLMs. First, in Section 3.1, we examine the long CoT trajectories generated by such models, verifying the common existence of intrinsic parallelism. Subsequently, Section 3.2 details probing and prompting tests, showing that AR-LLMs are unable to explicitly enforce or discern this parallelism during generation.

## 3.1 LLMs can Implicitly Generate Parallelizable Branches.

We start by analyzing the long CoT trajectories of current AR-LLMs using the s1K-1.1 dataset [27], including Deepseek R1 [16] and Gemini 2.0 Flash Thinking [13], aiming to answer the following question: *Does the logic of sequentially generated text truly depend on all content that precedes it?*

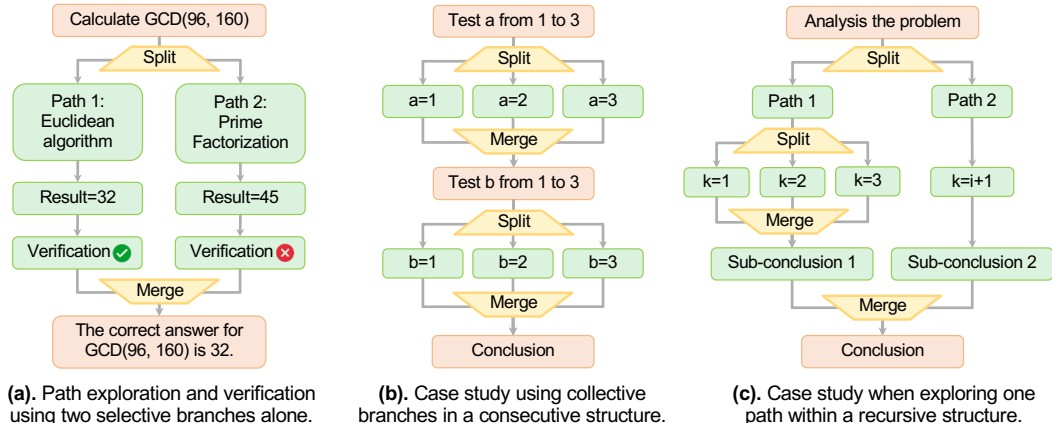

**(a).** Path exploration and verification using two selective branches alone.

**(b).** Case study using collective branches in a consecutive structure.

**(c).** Case study when exploring one path within a recursive structure.

Figure 2: Parallelizable branches are either collective (with all branches contributing) or selective (with some branches contributing). They can exist alone or as a part in consecutive/nested structures.

Table 1: Parallelizable branches commonly exist in long CoT trajectories generated by AR-LLMs. Per-example existence ratio (R%) and frequency (F) of different types are measured in the format R|F.

| | Collective Branch | | | Selective Branch | | Total |
|---|---|---|---|---|---|---|
| | Case Study | Subtask Execution | Other | Path Exploration | Other | |
| Deepseek R1 | 28.0 \| 2.00 | 47.3 \| 3.38 | 3.2 \| 0.23 | 19.4 \| 1.39 | 1.1 \| 0.07 | 99.0 \| 7.07 |
| Gemini 2.0 Flash | 39.4 \| 2.82 | 45.0 \| 3.22 | 2.9 \| 0.21 | 8.3 \| 0.60 | 1.4 \| 0.09 | 97.0 \| 6.94 |

Surprisingly, we observe distinct cases, termed "parallelizable branches", where multiple generation steps can be executed concurrently, rather than strictly awaiting the completion of prior text generation. These cases highlight the inherent parallelism in AR-LLMs. Figure 2 exemplifies these branches, which are categorized as collective and selective, that can be combined consecutively or recursively.

**Collective Branches.** This involves independent steps whose outputs are subsequently merged into the final result. Such scenarios often result from splitting a complex task into subtasks that can be processed concurrently. Examples include studying different cases and analyzing individual events.

**Selective Branches.** This refers to cases where numerous paths are considered, but not all contribute to the final output. Examples include exploring diverse solutions or examining competing hypotheses.

Table 1 further details the occurrence ratios and frequencies for different types in s1K-1.1, where over 98% of examples involve parallelizable branches. Among them, collective branches (e.g., case study and subtask execution) are dominant, accounting for 79%, with selective branches like path exploration comprising the other 19%. Moreover, these branches appear frequently, averaging 7 times per example, which are arranged into combinations of consecutive and recursive parallel structures.

## 3.2 LLMs cannot Explicitly Structure Parallelizable Branches.

Next, we examine the behavior of AR-LLMs from both token and hidden spaces, revealing their current inability to explicitly generate or identify such parallelizable branches based on our structures.

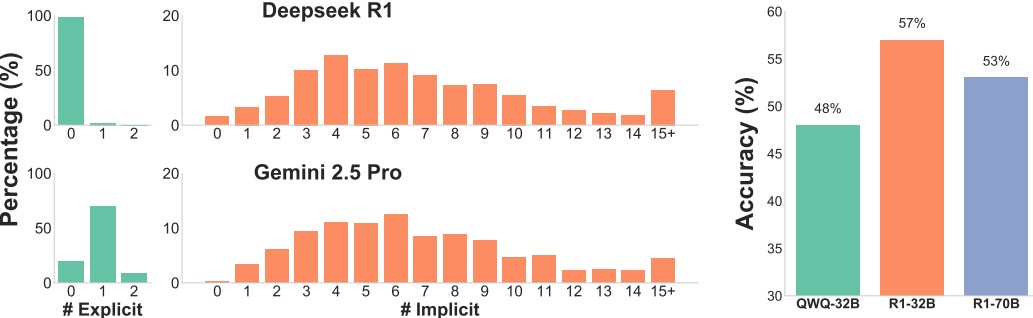

(a) Prompting Test: Comparing Explicit and Implicit Structure Counts.

(b) Probing Test: Classifier Acc.

Figure 3: Our tests show AR-LLMs cannot explicitly *enforce or discern* parallelizable branches.

**Prompting Test.** We first prompt Deepseek R1 and Gemini 2.5 Pro using the same questions, with a detailed description of parallel structures that includes both types and their combinations. As shown in Figure 3a, a significant 90% disparity emerges between explicit occurrence and implicit existence of these structures, highlighting that current LLMs are unable to perform explicit parallel generation.

**Probing Test.** Next, we delve deeper into the hidden space of AR-LLMs, conducting probing test to confirm whether they can discern intrinsic parallelism. Specifically, we label the tokens before each parallel block as positive examples and treat all other tokens as negative. Final-layer representations of these tokens are extracted using DeepSeek-R1-Distill-Qwen-32B & 70B [9] and QWQ-32B [34]. A two-layer MLP classifier is trained to predict whether a token initiates a parallel structure. However, the classifier's low test accuracy in Figure 3b suggests that AR-LLMs do not truly understand such parallelism but generate these structures unconsciously based on patterns from the pre-training corpus.

## 4 Designing Multiverse for Natively Parallel Generative Modeling.

With all findings in Section 3, we present Multiverse, a new generative modeling framework built on the MapReduce paradigm that explicitly decides how to parallelize and merge the generation process.

### 4.1 Preliminaries.

**Language Modeling** aims to learn the joint probability distribution over sequences of words or tokens. Given a finite vocabulary $V$ of tokens, and a sequence of $L$ tokens denoted by $\mathbf{x}^{1:L} = (x^1, x^2, \ldots, x^L)$, a language model estimates the joint probability $P(x^1, x^2, \ldots, x^L)$ of the sequence.

**Autoregressive Modeling** involves representing a sequence from left to right, where the probability of each token $x_t$ is conditioned on all previously tokens in the sequence (i.e., $\mathbf{x}^{1:t-1}$). Consequently, the joint probability of the entire sequence $\mathbf{x}^{1:L}$ is factorized as a product of conditional probabilities:

$$P(\mathbf{x}^{1:L}|\theta_{AR}) = P(x^1, x^2, \ldots, x^L|\theta_{AR}) = \prod_{t=1}^{L} P(x_t|x_1, \ldots, x_{t-1}; \theta_{AR})$$

where $\theta_{AR}$ denotes model parameters. AR models offer high accuracy but exhibit poor parallelism.

### 4.2 Multiverse Modeling.

Our modeling framework, Multiverse, advances beyond AR by eliminating redundant sequential dependencies between independent sequences, enabling adaptive and lossless parallel generative modeling. Therefore, our Multiverse must "smartly" decide when to start and end parallel generation. To realize this, we adopt a MapReduce structure internalizing three stages, as illustrated in Figure 4.

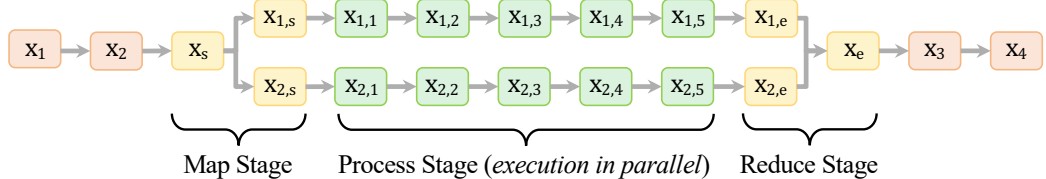

Figure 4: Multiverse enables natively parallel generation by internalizing a MapReduce paradigm.

**Map Stage.** The pipeline begins by generating a task decomposition plan, denoted as $\mathbf{x}_s$. Each subtask in $\mathbf{x}_s$ is then mapped to an independent prefix sequence, modeled as $P(\mathbf{x}_{1,s}|\mathbf{x}_s)$ and $P(\mathbf{x}_{2,s}|\mathbf{x}_s)$.

**Process Stage.** Next, it performs parallel modeling for each branch independently, conditioned on its own prefix. This enables the concurrent generation of diverse branches, like: $P(\mathbf{x}_{1,1:6}|\mathbf{x}_{[1:3,s]}, \mathbf{x}_{1,s})$ and $P(\mathbf{x}_{2,1:6}|\mathbf{x}_{2,s}, \mathbf{x}_{[1:3,s]})$. Each branch should end if a specific suffix (i.e, $\mathbf{x}_{1,e}$ or $\mathbf{x}_{2,e}$) is generated.

**Reduce Stage.** After completing all branches, Multiverse shift back to sequential generation that conditioned on all preceding tokens, which is modeled as $P(\mathbf{x}_{e,[3:4]}|\mathbf{x}_{1,[s,1:6,e]}, \mathbf{x}_{2,[s,1:6,e]}, \mathbf{x}_{[1:3,s]})$.

The integration of this three-stage pipeline enables Multiverse to: (i) adaptively decide when and how to parallelize generation during the Map stage; and (ii) retain informational completeness by ensuring every branch remains fully accessible throughout the Reduce stage and beyond. Notably, Multiverse naturally generalizes to both recursive and consecutive compositions of multiple MapReduce blocks.

### 4.3 Structured Generation Flow.

To enable automatic and interpretable control over the generation flow, Multiverse further employs a structured set of specialized control tags that explicitly define each MapReduce block. These tags, such as <Parallel>, delineate the boundaries of such blocks and coordinate all three internal stages. An example is provided in Figure 5.

The process begins with the **Map** stage, initiated by the <Goal> tag. This tag defines the overall objective, which is then broken down into subtasks using nested and indexed <Outline> tags. After goal specification (signaled by </Goal>), the **Process** stage starts. In this stage, each subtask is independently mapped and processed within a <Path> block in parallel, matched by its index, constituting the Process stage. Once all paths have finished (signaled by </Path>), the <Conclusion> tag triggers the **Reduce** stage that merges the results from these paths into a final coherent output ended with </Conclusion> tags.

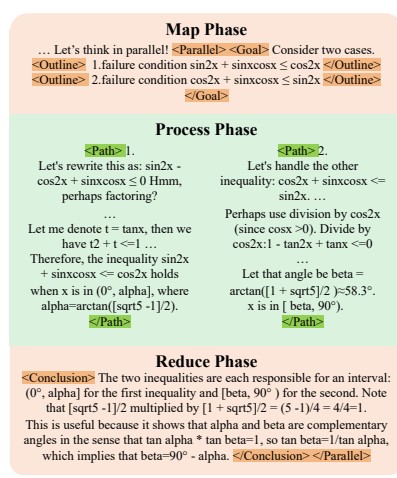

Figure 5: MapReduce Structure.

## 5 Building a Real-world Multiverse Reasoning Model.

To deploy Multiverse in real-world scenarios, we present a comprehensive suite consisting of *Multiverse Curator* as the data generator, *Multiverse Attention* as the model architecture, and *Multiverse Engine* as the serving system. This suite enables a seamless and rapid transition from leading AR models to Multiverse models. In particular, we apply this suite in complex reasoning tasks, leading to a Multiverse model that exhibits strong reasoning capabilities using remarkably low training cost.

### 5.1 Data Curation: Multiverse 1K.

To address the absence of MapReduce structures in existing sequential reasoning data, we introduce **Multiverse-1K**. While these long CoT trajectories often inherently contains such structures, explicitly generating them is difficult, as detailed in Section 3. Thus, we develop an automated LLM-assisted pipeline that transforms sequential reasoning chains into parallel MapReduce structures. This convert is guided by a five-stage prompting protocol powered by Gemini 2.5 Pro [14] as shown in Figure 6a.

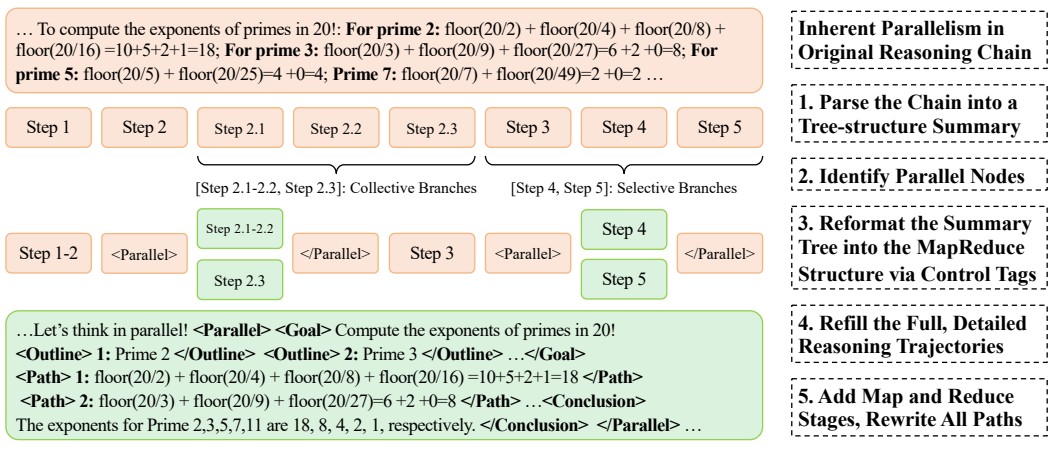

(a) **Multiverse Curator** automatically generated **Multiverse-1K** using an LLM-assisted data curation pipeline.

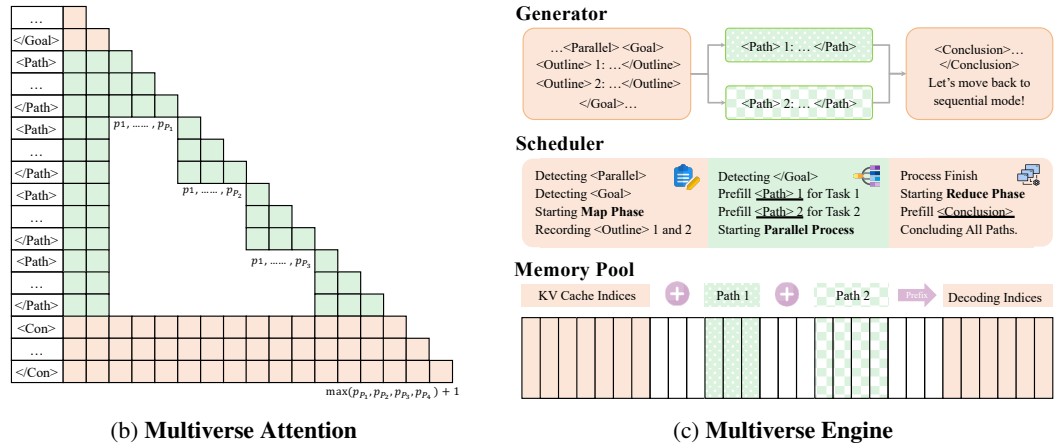

(b) **Multiverse Attention**    (c) **Multiverse Engine**

Figure 6: Multiverse co-design data (*Multiverse 1K*), algorithm (*Multiverse Attention*), and system (*Multiverse Engine*) to enable real-world reasoning capabilities through a rapid shift from AR-LLMs.

**Generating a Summary Tree.** First, we iteratively decompose and outline the original reasoning chain into a two-level tree structure. In the first round, the entire reasoning chain is broken down into multiple steps. In the second round, each step is examined by the LLM for further decomposition into substeps. Each resulting step or substep will be labeled and outlined with a concise description.

**Identifying Parallel Groups.** Second, we instruct the LLM to analyze each reasoning step, identifying which steps or groups of steps can be executed in parallel without violating logical dependencies.

**Reformating into Parallel Structures.** Third, the summary tree is converted into a parallel structure based on the previous analysis. To explicitly signal parallel execution, parallelizable steps or step groups are enclosed within the control tags `<Parallel>` and `</Parallel>`, forming a parallel block.

**Refilling Original Details.** Fourth, we prompt the LLM to repopulate the detailed content for each step and substep. This is achieved by retrieving and copying the related original reasoning trajectories.

**Adding MapReduce Structures**. Finally, we further convert the parallel structures into MapReduce structures as defined in Section 4.3. For each parallel block, the LLM generates both the Map and Reduce stages by outlining the specific goals and results for each individual path. Moreover, all paths are rewritten to avoid words implying sequential relations (e.g., similarly) and to prevent including or referencing content from other paths, thereby ensuring each path's completeness and independence.

To further refine our data, two supplementary validation stages are incorporated. After the fourth stage, a content check will filter out data if its edit distance ratio is above 0.2. Next, after the fifth

stage, a grammar check will confirm strict adherence to our MapReduce structures. Data failing either case will be iteratively regenerated through our pipeline until both standards are met. The application of this automated pipeline to the s1K-1.1 dataset has yielded Multiverse 1K, a new dataset consisting of 1,000 high-quality, structured reasoning trajectories across a range of math and science problems.

## 5.2 Algorithm Design: Multiverse Attention.

Next, we introduce **Multiverse Attention** to replace the causal attention [39] in AR-LLMs. Causal attention computes the $i$-th token's output with query $q_i$, and keys $k_j$, values $v_j$ from positions $j \leq i$:

$$a_{ij} = \text{Softmax}\left((q_i^\top + P(i)) \cdot (k_j + P(j)) + M_{ij}\right),$$ (1)

where $M_{ij} = \begin{cases} 0, & j \leq i \\ -\infty, & \text{otherwise} \end{cases}$ is the causal mask, and $P(i)$ is the embedding for the $i$-th position.

However, this formulation poses challenges for parallel generation, as the computation of later paths depend on both (i) the key-value (KV) pairs and (ii) the positional indices produced by earlier paths. To address this issue, we modify both the attention masks and position indices following APE [44], as illustrated in Figure 6b. In Multiverse Attention, each path within the same `Process` block starts from an identical position and executes independently without accessing others. During the `Reduce` stage, all paths converge to the same position, which is set to the maximum position reached by any path to ensure non-negative relative distances, while accommodating variable-length paths.

Moreover, its structural similarity to causal attention brings two key efficiency benefits: (i) **Hardware Efficiency:** it can preserve training parallelism by using customized attention kernel in FlexAttention [43], and (ii) **Data Efficiency:** it can be rapidly adapted via post-training on a few samples.

## 5.3 System Implementation: Multiverse Engine.

To enable truly parallel generation in practical deployments, we introduce **Multiverse Engine**, an extension of existing inference engines designed for AR models. Specifically, we employ SGLang [50] due to its support for continuous batching and radix attention. These features allow dynamic batch scheduling and flexible KV-cache reuse, two scenarios frequently occur in the `Map` and `Reduce` stages.

The `Map` stage is automatically triggered when a `<Parallel>` token is generated. Next, the scheduler counts the number of `<Outline>` encountered to decide the degree of parallelism until reaching `</Goal>`. Based on this count, the engine creates multiple paths executed in parallel as distinct samples within the same batch. Leveraging radix attention, these paths share the prefix KV cache from the current context. Each path is identified and initiated with "`<Path> i`" according to its order `i` in the `<Outline>` list. After prefilling, all paths are added to the decoding queue for parallel generation. When a path finishes, either by reaching `</Path>` or the maximum length, it enters a "zombie" state that releases all resources and waits for the completion of other paths before continuing.

The `Reduce` stage begins once all processing paths have completed. In this stage, the engine merges the KV states from all paths along with the preceding context to form a new sequence. Thanks to the flexible memory layout of the radix cache, indices of KV cache can be seamlessly merged without any padding, thereby avoiding both physical data copying and subsequent redundant computation. The token `<Conclusion>`, prefixed by this combined KV cache, is then added to the prefilling queue. Once finished, the task is moved to the decoding queue to resume generation along the new sequence.

# 6 Experiments

We evaluate the effectiveness and efficiency of Multiverse in real-world reasoning tasks. Specifically,

- In Section 6.2, Multiverse-32B achieves substantial gains over the Qwen2.5 model by 23.6% after training on Multiverse-1K, and matches or exceeds the accuracies of AR-LLMs on reasoning tasks.
- In Section 6.3, Multiverse-32B scales better than AR-LLMs when using the same generation length.

## 6.1 Setup.

**Training.** We created Multiverse-32B by performing SFT on the Qwen2.5-32B-Instruct model [33], integrating our Multiverse Attention. The training data consisted of a combination of Multiverse 1K

Table 2: Performance comparison between Multiverse-32B and other 32B autoregressive LLMs. The pass@1 metric is reported using LightEval [17], while # parallel computes the ratio between the total number of generated tokens and the actual generation length, measuring the degree of parallelism.

| Model / Metric | AIME24 | | AIME25 | | MATH500 | | GPQA-Diamond | |
|---|---|---|---|---|---|---|---|---|
| | pass@1 | # parallel | pass@1 | # parallel | pass@1 | # parallel | pass@1 | # parallel |
| s1-32B | 35.4 | 1.00 | 25.8 | 1.00 | 88.6 | 1.00 | 48.0 | 1.00 |
| s1.1-32B | 52.9 | 1.00 | 41.7 | 1.00 | 93.4 | 1.00 | 60.3 | 1.00 |
| Qwen2.5-32B-Instruct | 15.8 | 1.00 | 10.4 | 1.00 | 80.4 | 1.00 | 47.0 | 1.00 |
| Autoregressive-32B | 51.3 | 1.00 | **42.9** | 1.0 | **92.8** | 1.00 | 61.6 | 1.00 |
| **Multiverse-32B-zero** | 52.1 | 1.07 | 44.2 | 1.05 | 91.8 | 1.05 | **62.1** | 1.06 |
| **Multiverse-32B** | **52.9** | 1.24 | 44.2 | 1.18 | 92.4 | 1.15 | 61.7 | 1.17 |

prompted with "Think step by step and in parallel", and the original sequential data appended by "Think step by step", using a mixture ratio increased from 0:1 (all original data) to 1:0 (all our data) across eight epochs. Our fine-tuning took 3 hours on 8 NVIDIA B200 GPUs with PyTorch FSDP.

**Evaluation.** Following common practice in assessing reasoning models, we measure Multiverse-32B on four tasks, including AIME24 [24], AIME25 [25], MATH500 [18], and GPQA Diamond [35]. LightEval [17] is employed as the evaluation toolkit, powered by our SGLang [50]-based Multiverse Engine. We test our model under two prompting conditions: with and without the phrase "in parallel".

**Baselines.** We compare our model with the Qwen2.5 model and an Autoregressive-32B trained using the same data, but without any control tags or the Map and Reduce stages. In addition to pass@1, we report the degree of parallelism (# Parallel) as the ratio between generated token counts and lengths.

## 6.2 Real-world Reasoning Performance

In Table 2, we report the performance of Multiverse-32B on complex reasoning tasks with 32K contexts, showing improvements of 36%, 32%, 12%, and 15% over the Qwen2.5-32B-Instruct model across the respective benchmarks after fine-tuning. Notably, Multiverse-32B matches or even surpasses the performance of autoregressive models, as demonstrated by its comparison with Autoregressive-32B. For reference, we also include the results of the s1.1-32B model trained on the sequential CoT data from which Multiverse-1K is derived. The comparable performance between these models confirms that our data curation pipeline successfully preserves the original data quality.

We also evaluate Multiverse-32B-Zero, a variant prompted without the "think in parallel" instruction. While both versions exhibit similar performance, Multiverse-32B achieves greater parallelism. This parallelism, measured as the ratio of generated tokens to generation length, aligns with our training strategy, suggesting the potential for controllably switching between AR and Multiverse generation. Moreover, the reduced parallelism on AIME tasks indicates that the model exhibits less parallelism during longer generation, partly due to the scarcity of data exceeding 16K tokens in Multiverse-1K.

## 6.3 Scaling Performance

To highlight the benefits of parallel generation, we conduct budget control experiments on GPQA-Diamond and MATH500 using the same context length (i.e., approximately equal generation time). We vary the context length from 1K to 4K tokens, during which accuracy increases sharply. As shown in Figure 7, while longer contexts improved performance for both models, Multiverse-32B generates more tokens within the same context length. This parallel scaling yielded performance gains of 2.23% on the GPQA-Diamond (with # parallel = 1.17) and 1.51% on the MATH500 (with # parallel = 1.15).

## 7 Efficiency Analysis

Having demonstrated Multiverse-32B's strong scalability and overall performance, we now further analyze the practical efficiency of Multiverse, showing the potential unlocked through parallel scaling.

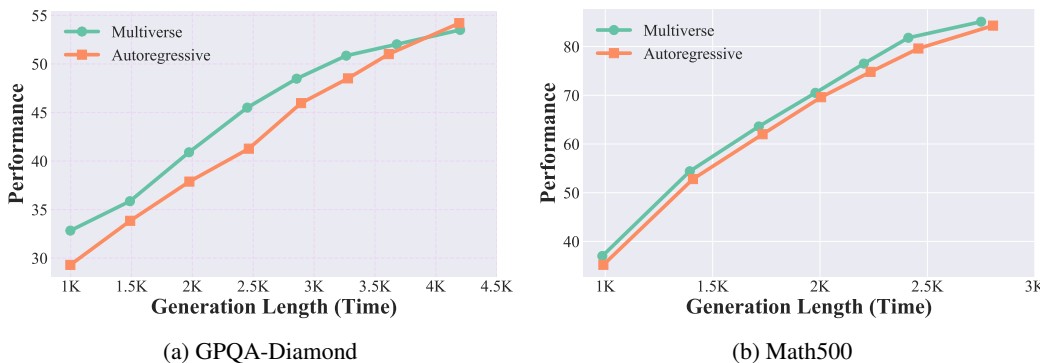

| (a) GPQA-Diamond | (b) Math500 |

Figure 7: Multiverse achieves better performance using a fixed context length, indicating the same generation time. Due to parallel scaling, it generates more tokens within the same time. Here, we report the actual generation length, as some examples terminate before reaching the maximum length.

First, we investigate the relationship between the degree of parallelism and latency per token across various generation lengths (8K, 16K, and 32K), using a batch size of one. The resulting data points, illustrated in Figure 8a, demonstrate that Multiverse enhances generation efficiency by increasing the degree of parallelism. Furthermore, we fit the sampled data points into three inverse curves, one for each. These curves highlight the potential of Multiverse to further reduce latency by encouraging parallelism. Specifically, we identify three key regions based on the sample distribution, demarcated by red lines. The first, encompassing parallelism degrees from 1.0 to 1.3, represents the majority of data points and reflects real-world scenarios, yielding an average speedup of 18.5%. Furthermore, examples show that higher parallelization is achievable, offering acceleration up to 2.1×. The final region, characterized by extended lines, indicates the promising potential for further improvements.

Next, we show the speedup achieved by Multiverse-32B with varying degrees of parallelism across different batch sizes, while keeping a fixed 4K output length. The results in Figure 8b indicate that inference remains memory-bound as the batch size increases from 1 to 128. Therefore, the speedup of Multiverse is influenced by the degree of parallelism in multiple scenarios, showcasing its scalability.

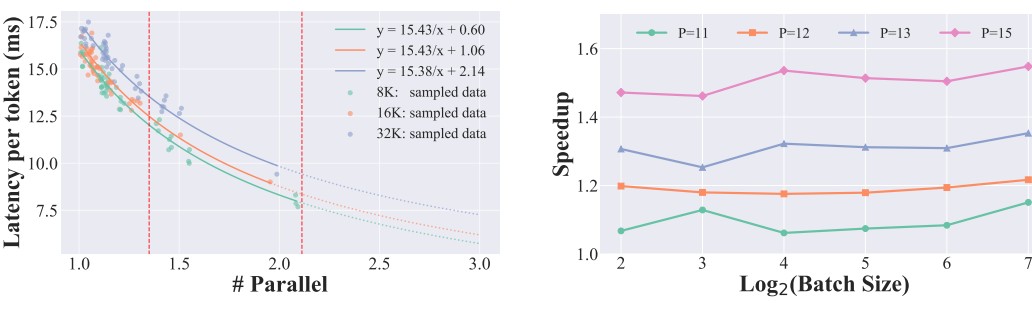

| (a) Reduced Latency/Token with Increased # Parallel | (b) Stable Speedup Across Varying Batch Size |

Figure 8: Multiverse can obtain scalable efficiency gain based on the degree of parallelism.

## 8  Conclusion

This work proposes Multiverse, a natively parallel generative model based on a MapReduce paradigm that internalizes three stages: (i) a `Map` stage for adaptive task decomposition, (ii) a `Process` stage for parallel subtask execution, and (iii) a `Reduce` stage for lossless result synthesis. To build a real-world Multiverse reasoning model, we co-design our data, algorithm, and system, enabling a seamless and rapid transfer from AR-LLMs. After fine-tuning on Multiverse-1K, our Multiverse-32B achieves performance comparable to AR-LLMs on real-world reasoning tasks, while achieving better performance using the same context length due to parallel scaling. Additionally, such parallel generation also results in an up to 2× efficiency gain across varying batch sizes, based on the degrees of parallelism. We hope Multiverse can be considered a successor to Autoregression for generative modeling. For the Limitations and Broader Impacts, we will discuss them in detail in our Appendix.

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

## Acknowledgement

We thank Zhuoming Chen, Haizhong Zheng, Ranajoy Sadhukhan, Yang Zhou, Songlin Yang, Liliang Ren, Wentao Guo, Ruijie Zhu, Yu Zhang, and Yixin Dong for their constructive feedback on this work, along with the authors of s1 [27], SGLang [50], and LightEval [17] for their useful codebase. We are also grateful to BitDeer AI Research for providing GPU resources and to Google for supplying API credits. This work was supported in part by a DGX B200 gift from NVIDIA, a Google Research Award, an Amazon Research Award, Intel, Li Auto, Moffett AI, and the CMU CyLab Seed Fund.

## A    Limitations

While Multiverse provides a general framework for generative modeling, its application to diverse data and task types beyond LLM reasoning remains underexplored. Moreover, as Multiverse-32B was trained solely using Supervised Fine-Tuning (SFT), a key direction in future research is to integrate Reinforcement Learning (RL) into training to explore and encourage more parallelism, which in turn would require a more robust Multiverse engine.

## B    Broader Impacts

Multiverse significantly boosts GPU utilization by enabling massive parallel generation. This modeling framework is particularly beneficial for small-batch and long-context inference scenarios, leading to substantial reductions in latency and corresponding energy consumption. Furthermore, Multiverse enables economies of scale for difficult but parallelizable tasks, decreasing the time per task unit while maintaining near-constant overall latency, even as task complexity increases. This remarkable scalability showcases its potential to address extremely complex tasks in practice that were previously intractable, offering a promising path towards artificial superintelligence (ASI).

## C    Prompt of Multiverse Curator

In this section, we release our complete five-stage prompting protocol to create Multiverse-1K, powered by the Gemini 2.5 Pro model. This protocol is engineered to transform any sequential CoT data into Multiverse data.

This protocol starts with a multi-round conversation with the LLM (Stages 1-3) to convert an original reasoning chain into a parallel-structured summary. In Stage 4, both this summary and the original reasoning trajectory are fed to the LLM to repopulate each summarized step with its complete, original details. A content checker then immediately assesses these refilled steps. If the *editor distance* (e.g., Levenshtein distance between the original trajectory ($s_{ori}$) and its rewritten version ($s_{gen}$), denoted as $d(s_{ori}, s_{gen})$) is too high, that step is re-generated. To normalize this, a *relative editor distance* is calculated to decide if a threshold $r$ is exceeded (set to 0.2 in practice):

$$\text{Relative Editor Distance} = \frac{d(s_{ori}, s_{gen})}{\max(\text{length}(s_{ori}), \text{length}(s_{gen}))}$$

Next, in Stage 5, we transform the output from Stage 4 into a MapReduce-structured reasoning trajectory by inserting the Map and Reduce phases that are generated by Gemini 2.5 Pro. To ensure the structural validity of the data, we perform a grammar check using a customized XML interpreter, which filters out invalid entries and extracts the outermost MapReduce blocks in the remaining valid ones. Finally, each path is rewritten separately to produce fully independent reasoning paths. The prompts used in the entire protocol are as follows:

## STAGE 1: Generating a Summary Tree

**Main-Step Extraction**

Analyze the given reasoning chain (for a math or coding problem) and pull out every **major** step. Ignore substeps—only list the top-level insights or actions.

*Output format*

```
S1: [First major step]
S2: [Second major step]
S3: [Third major step]
...
SX: [Description of step X]
...
```

*Guidelines*

- Label each top-level step consecutively ('S1', 'S2', 'S3', ...).
- Please capture the entire thought process presented in the reasoning chain, and do not skip any step that includes but not is limited to:
    1. Initial problem understanding and analysis
    2. All exploration paths (both successful and unsuccessful)
    3. Case studies, checks, or tests performed
    4. Any "aha" or correction (re-evaluation or re-thinking) moments
    5. The final reasoning that yields the solution
- Keep each item concise yet descriptive.
- Do **not** include any sub-numbering (no 'S2.1', etc.).
- Explicitly split multiple cases or scenarios into different steps. Each case should be allocated an independent step.

**Substep Extraction**

Given the output including all main step from a reasoning chain, break it down into all its internal substeps only if it can be meaningfully subdivided into smaller thought units.

*Output format*

```
S1: [Description of step 1]
S2: [Description of step 2]
    S2.1 [Description of step 2.1]
    S2.2 [Description of step 2.2]
    ...
    S2.10 [Description of step 2.10]
S3: [Description of step 3]
S4: [Description of step 4]
...
S10: [Description of step 10]
...
```

*Guidelines*

- Use the same parent index ('x') as the main step (e.g. if breaking down 'S2', label 'S2.1', 'S2.2', ...).
- Capture the entire thought process presented in the reasoning chain, and do not skip any substep that includes but is not limited to:
    1. Initial problem understanding and analysis
    2. All exploration paths (both successful and unsuccessful)
    3. Case studies, checks, or tests performed
    4. Any "aha" or correction (re-evaluation or re-thinking) moments
    5. The final reasoning that yields the solution
- Do **not** introduce deeper nesting larger than 2 (e.g. 'S2.1.1' is not allowed).
- Explicitly split multiple cases or scenarios into different substeps. Each case should be allocated an independent substep.

## Parallelizing Main Steps

Using only the **main steps** (S1, S2, . . . ) you extracted in Stage 1, identify all steps or contiguous step groups that can be executed in parallel without violating logical dependencies, and rewrite the plan as a structured parallel execution outline.

*1. Identify Parallel Groups*
- Find sets of adjacent main steps with no dependencies among them.
- Label groups P1, P2, . . . and list their step ranges (e.g. [S1+S2, S3], [S4]).

*2. Rewrite into a Parallel Execution Plan*
- Preserve each step's original wording as much as possible.

*Output Format:*
```
Parallel groups:
P1: [S1+S2, S3]
P2: [S4]
...

Parallel execution plan:
P1[parallel reason: ...]:
  S1+S2: [text of S1 + text of S2]
  S3: [text of S3]
P2[parallel reason: ...]:
  S4: [text of S4]
...
```

*Guidelines*
- **Coverage**: Include **every** step exactly once, either alone or inside a parallel group.
- **Contiguous Blocks**: Combine only adjacent steps into blocks; do **not** combine non-adjacent steps.
- **Strict Parallelism Only**: Build a dependency graph: draw an edge from step A to B if B uses A's output. A group P_i may include steps (or blocks) only if there are no edges between them. Treat conditional branches as independent tasks.
- **Contiguous Grouping Only**: Each parallel group must cover a continuous sequence of steps. Do not parallelize non-adjacent steps.
- **Conciseness**: Keep each bullet short and stick closely to the original text.

## Parallelizing Substeps

Using only the **substeps** (S2.1, S2.2, ...) you extracted in Stage 1, identify all substeps or contiguous substep groups can be executed in parallel without violating logical dependencies, and rewrite the plan as a structured parallel execution outline.

*1. Identify Parallel Groups*
- Find sets of adjacent main steps with no dependencies among them.
- Label groups P1, P2, . . . and list their step ranges (e.g. [S2.1+S2.2, S2.3], [S3.1]).

*2. Rewrite into a Parallel Execution Plan*
- Preserve each step's original wording as much as possible.

*Output Format:*
```
Parallel groups:
P1: [S2.1+S2.2, S2.3]
P2: [S2.4]
P2: [S3.1]
...

Parallel execution plan:
P1[parallel reason: ...]:
  S2.1+S2.2: [text of S2.1 + text of S2.2]
  S2.3: [text of S2.3]
P2[parallel reason: ...]:
  S3.1: [text of S3.1]
```

...

*Guidelines*
- **Coverage**: Include **every** substep exactly once, either alone or inside a parallel group.
- **Contiguous Blocks**: Combine only adjacent substeps into blocks; do **not** combine non-adjacent substeps.
- **Strict Parallelism Only**: Build an explicit dependency graph in your analysis: draw an edge from substep A to substep B if B uses A's output or insight. A group `Pi` may include steps (or contiguous blocks) only if there are no edges between any two steps. In conditional logic, treat the **if** branch and **else** branch as independent tasks and parallelize them even though their outputs cannot both occur at runtime.
- **Contiguous Grouping Only**: Each parallel group must cover a continuous sequence of steps or blocks. In other words, you may only parallelize adjacent substeps. The occurrence of substeps in parallel groups must follow their original order. For example, P1: `[S2.2, S3.1]` is not allowed.
- **Conciseness**: Keep each bullet short and stick closely to the original text.

## STAGE 3: Reformating into Parallel Structures

**Get Structured Summary**
Please summarize the conversation above by extracting the reasoning steps and substeps in Stage 1 as a tree structure with explicit parallelism annotations following Stage 2.

*Output Format*
```
O1: [Brief summary of top-level step S1]
<parallel>[parallel reason: ...]
O1.1: [Summary of substep S1.1]
O1.2: [Summary of substep S1.2]
...
</parallel>
<parallel>[parallel reason: ...]
O2: [Brief description of top-level step S2]
<parallel>[parallel reason: ...]
O2.1: [Summary of substep S2.1 + Summary of substep S2.2]
O2.2: [Summary of substep S2.3]
...
</parallel>
O3: [Brief description of top-level step S3]
</parallel>
O4: [Brief description of top-level step S4]
...
```

*Guidelines*
- **Max depth of nested `<parallel>` is 2.** Do not nest <parallel> tags more deeply than two levels.
- **Max depth of nested numbering is 2.** Only use `Ox` and `Ox.y`; do not introduce deeper numbering like `Ox.y.z`.
- **Sequential subpaths stay unexpanded.** If a node's children are purely sequential, list them normally without any <parallel> wrapper.
- **Tag parallel blocks.** Wrap only genuinely parallelizable sibling steps in a <parallel>...</parallel> block, and include a parallel-reason annotation.
- **Concise summaries.** Each step and substep should be described briefly and clearly.
- **Avoid over-splitting.** If most children are sequential and only a pair can run in parallel, either leave the group un-split or tag only the truly parallel pair.
- **Group parallelizable sets.** You may combine several independent paths into one <parallel> block when they share no dependencies.

## STAGE 4: Refilling Original Details

**Refill the Full, Detailed Reasoning Trajectories into the Structured Summary**

You will receive an outline that *may be incomplete but includes* `<parallel>` *tags* indicating parallel structures. It contain summaries for several steps and substeps. You will also receive the corresponding original text, where sentences implicitly or explicitly map to hierarchical prefixes (e.g., O1, O1.1, O2) in sequence. Your task is to process the original reasoning chain sequentially to update the outline: replace existing summaries or insert new steps as needed, while preserving the original `<parallel>` tag structure.

*Guidelines:*

- **Initialize Structure** Start with the structure provided by the input outline, including its text/summaries and all `<parallel>` tags in their original locations.
- **Read Sentences Sequentially:** Process each sentence of the original text one by one, in the exact order they appear.
- **Process Each Sentence:**
  1. Determine the hierarchical prefix associated with this sentence (e.g., O1, O1.1, O2).
  2. Check if a step or substep with this prefix already exists in the outline.
  3. *If it exists:* Replace its current summary with the full original sentence.
  4. *If it does not exist:* Insert a new step/substep at the correct hierarchical position (e.g., S1.1 under S1, S2 after S1), using the full original sentence as its content and matching the outline's indentation.
- **Preserve `<parallel>` Tags:** Keep every existing `<parallel>` and `</parallel>` tag exactly where it was in the input outline. Do not add, remove, or relocate any tags.
- **Ensure Correct Output Formatting:**
  - Maintain proper hierarchical indentation for all steps and substeps.
  - Each entry must be on its own line, beginning with its prefix (e.g., O1:, O1.1:), followed by the full original sentence.
- **Maintain Completeness:** Verify that every sentence from the original reasoning chain has been processed and appears in the updated outline. Do not omit or merge any sentences.

## STAGE 5: Adding MapReduce Structures & Rewriting All Paths

**Filling Detailed Goal and Conclusion Based on the New Reasoning Trajectory**

Based on the generated reasoning chain, your task is to transform it according to the following rules:

*Output Format*

```
[Full reasoning copied from the reasoning chain for the first top-level path]
[Full reasoning copied from the reasoning chain for the second top-level path]
...
Let's think in parallel.
<Parallel>
<Goal>
Path: [brief, self-contained description of case A]
Path: [brief, self-contained description of case B]
...
</Goal>
<Path>
[Introductory reasoning for case A]
Let's think in parallel.
<Parallel>
<Goal>
Path: [brief, self-contained description of case A.1]
Path: [brief, self-contained description of case A.2]
</Goal>
<Path>
[Full detailed reasoning for case A.1, rewritten clearly and independently]
</Path>
```

```
<Path>
[Full detailed reasoning for case A.2, rewritten clearly and independently]
</Path>
<Conclusion>
[Your concise summary of outcomes from A.1 and A.2]
</Conclusion>
</Parallel>
</Path>
<Path>
[Full detailed reasoning for case B, rewritten clearly and independently]
</Path>
...
<Conclusion>
[Your concise summary of outcomes from A and B]
</Conclusion>
</Parallel>
[Full detailed reasoning for any remaining paths]
```

*Guidelines*
- Remove all numbering labels (e.g., `01`, `02.1`) and eliminate any indentation.
- For each `<Parallel>...</Parallel>` block:
  - Group every step, substep, and subsubstep belonging to the same parallel branch into a single `<Path>...</Path>` section.
  - Discard the `[parallel reason: ...]` annotations.
- Within each `<Parallel>` block:
  - Insert `<Goal>` before the first `<Path>`, listing each branch as `Path: ...` .
  - Insert `<Conclusion>` after the last `<Path>`, summarizing each branch's outcome independently.
- When multiple `<Path>` entries stem from the same original sentence or have interdependencies:
  - Rewrite each path separately and completely, ensuring no cross-references.
  - Provide enough context in each `<Path>` so it stands alone.
  - Fully encapsulate the logical reasoning for each path.
- Avoid repetition: do not echo the brief descriptions from `<Goal>` inside the corresponding `<Path>`, and minimize redundant information across paths.

## Rewriting Paths in the Structured Reasoning Trajectory

You are given a full structured reasoning trajectory inside a `<Parallel>` block, consisting of:
- one `<Goal>` block with multiple `<Outline>` elements
- multiple `<Path>` blocks
- one `<Conclusion>` block.

Some `<Path>` blocks may contain an entire nested `<Parallel>` structure (from `<Parallel>` to `</Parallel>`). These nested blocks should be rewritten using the same rules recursively.

**For `<Goal>`:**
- Rewrite each `<Outline>` into a **concise statement of what is being calculated or determined**.
- Remove any content describing **how** the problem is solved or intermediate reasoning steps.

**For each `<Path>`:**
- Keep the original numbering prefix (e.g., '1:', '2:').
- Rewrite the content as a **complete, fluent, and logically self-contained paragraph**.
- Do **not** use transitional phrases like "First," "Then," "Next," "On the other hand," etc.
- If the `<Path>` contains **five or fewer sentences**, rewrite them together as a single coherent paragraph, ensuring logical flow and fluency without using transitional phrases.
- If the `<Path>` contains **more than five sentences**: Rewrite the first five sentences together as a single unit, forming a fluent paragraph. For the remaining sentences, rewrite each one individually, based on its meaning, as clear and fluent standalone statements.
- If the `<Path>` contains a **nested `<Parallel>` block**, apply all these rules recursively to the nested block.

Each <Path> must make sense independently, even if it contains a nested reasoning chain.

**For <Conclusion>:**
- Rewrite the conclusion as the **most concise and synthesized summary** of the main outcomes from all <Path> blocks.
- You may combine or compare results from different paths, but keep it succinct and direct.

**Nested <Parallel>:**
- A nested <Parallel> may appear only **as a full block inside a <Path>**.
- If a <Path> contains a nested <Parallel>...</Parallel> block, process that inner block exactly as you would the top-level one:
  - Rewrite the inner <Goal>, <Path>, and <Conclusion> elements accordingly.
  - Maintain the XML structure — do not reindent or alter the tag hierarchy.

*Output Format*

```
<Parallel>
<Goal>
<Outline>
1: [concise description of the goal of Path 1]
</Outline>
<Outline>
2: [concise description of the goal of Path 2]
</Outline>
</Goal>
<Path>
1: [self-contained paragraph for Path 1, rewritten sentence by sentence]
<Parallel>
<Goal>
<Outline>
1.1: [concise goal for Path 1.1]
</Outline>
<Outline>
1.2: [concise goal for Path 1.2]
</Outline>
</Goal>
<Path>
1.1: [rewritten paragraph for Path 1.1]
</Path>
<Path>
1.2: [rewritten paragraph for Path 1.2]
</Path>
<Conclusion>
[summarized results from Paths 1.1 and 1.2]
</Conclusion>
</Parallel>
</Path>
<Path>
2: [self-contained paragraph for Path 2, rewritten sentence by sentence]
</Path>
<Conclusion>
[summary of outcomes from Path 1 and Path 2]
</Conclusion>
</Parallel>
```

*Guidelines*
- You must preserve all XML tags exactly as they appear — do not remove, rename, or reorder tags.
- Only modify the inner text of <Outline>, <Path>, and <Conclusion> blocks.
- Wrap the final output in a single Markdown code block using triple backticks.

# D    Examples in Training Data and Model Generation

We present several examples from our Multiverse-1K dataset, along with the corresponding outputs generated by Multiverse-32B, which was trained on this dataset. Both typical collective and selective branches are included.

## D.1    Training Examples from Multiverse-1K

---

**Multiverse Data A: Collective Case**

```
<Parallel>
<Goal>
<Outline>
```
1: Determine the sum of distances from $P_1$ to $P_2$, $P_3$, and $P_4$.
```
</Outline>
<Outline>
```
2: Determine the sum of distances from $P_2$ to $P_1$, $P_3$, and $P_4$.
```
</Outline>
<Outline>
```
3: Determine the sum of distances from $P_3$ to $P_1$, $P_2$, and $P_4$.
```
</Outline>
<Outline>
```
4: Determine the sum of distances from $P_4$ to $P_1$, $P_2$, and $P_3$.
```
</Outline>
</Goal>
<Path>
```
1: For point $P_1(0,0)$, the distance to $P_2(10,20)$ is

$$\sqrt{(10-0)^2 + (20-0)^2} = \sqrt{100 + 400} = \sqrt{500} \approx 22.36.$$

The distance to $P_3(5,15)$ is

$$\sqrt{(5-0)^2 + (15-0)^2} = \sqrt{25 + 225} = \sqrt{250} \approx 15.81.$$

The distance to $P_4(12,-6)$ is

$$\sqrt{(12-0)^2 + (-6-0)^2} = \sqrt{144 + 36} = \sqrt{180} \approx 13.42.$$

The sum is $22.36 + 15.81 + 13.42 \approx 51.59$.
```
</Path>
<Path>
```
2: For point $P_2(10,20)$, the distance to $P_1(0,0)$ is

$$\sqrt{(10-0)^2 + (20-0)^2} = \sqrt{500} \approx 22.36.$$

The distance to $P_3(5,15)$ is

$$\sqrt{(10-5)^2 + (20-15)^2} = \sqrt{25 + 25} = \sqrt{50} \approx 7.07.$$

The distance to $P_4(12,-6)$ is

$$\sqrt{(10-12)^2 + (20-(-6))^2} = \sqrt{4 + 676} = \sqrt{680} \approx 26.08.$$

The sum is $22.36 + 7.07 + 26.08 \approx 55.51$.
```
</Path>
<Path>
```
3: For point $P_3(5,15)$, the distance to $P_1(0,0)$ is

$$\sqrt{(5-0)^2 + (15-0)^2} = \sqrt{250} \approx 15.81.$$

The distance to $P_2(10,20)$ is

$$\sqrt{(5-10)^2 + (15-20)^2} = \sqrt{50} \approx 7.07.$$

The distance to $P_4(12,-6)$ is

$$\sqrt{(5-12)^2 + (15-(-6))^2} = \sqrt{49 + 441} = \sqrt{490} \approx 22.14.$$

The sum is $15.81 + 7.07 + 22.14 \approx 45.02$.

---

```
</Path>
<Path>
```
4: For point $P_4(12, -6)$, the distance to $P_1(0, 0)$ is

$$\sqrt{(12-0)^2 + (-6-0)^2} = \sqrt{180} \approx 13.42.$$

The distance to $P_2(10, 20)$ is

$$\sqrt{(12-10)^2 + (-6-20)^2} = \sqrt{4+676} = \sqrt{680} \approx 26.08.$$

The distance to $P_3(5, 15)$ is

$$\sqrt{(12-5)^2 + (-6-15)^2} = \sqrt{49+441} = \sqrt{490} \approx 22.14.$$

The sum is $13.42 + 26.08 + 22.14 \approx 61.64$.
```
</Path>
<Conclusion>
```
The sums of distances from $P_1, P_2, P_3, P_4$ to the other three points are approximately $51.59$, $55.51$, $45.02$, and $61.64$, respectively.
```
</Conclusion>
</Parallel>
```

---

## Multiverse Data B: Selective Case

```
<Parallel>
<Goal>
<Outline>
```
1: Determine if $H$ can be constructed as a countable intersection of open dense sets using Baire Category Theorem (BCT), leveraging the completeness of $S^2$.
```
</Outline>
<Outline>
```
2: Determine if $H$ can be constructed step-by-step using $\Omega$'s approximation property to ensure density and $K$-avoidance for chords, while addressing potential difficulties.
```
</Outline>
</Goal>
<Path>
```
1: The sphere $S^2$ is a complete metric space, suggesting the possibility of constructing $H$ as a countable intersection of open dense sets; however, the specific method for such a construction is not immediately clear.
```
</Path>
<Path>
```
2: A step-by-step construction of $H$ could leverage the given condition to ensure density and that all chords avoid $K$; $H$ must be dense in $S^2$, meaning it has points in every open subset of $S^2$, and any two points in $H$ must have their connecting chord disjoint from $K$. The family $\Omega$ allows approximation of any two points on $S^2$ with a chord $X'Y'$ from $\Omega$. This implies $\Omega$ is dense in the space of all chords, with the additional property that its chords avoid $K$. The closed nature of $K$ means its complement within the unit ball is open, so a chord disjoint from $K$ lies entirely within this open complement. Consequently, $\Omega$ comprises chords in the complement of $K$. The challenge is to construct a dense set $H$ such that all chords connecting points within $H$ avoid $K$, given that for any $X, Y \in S^2$, nearby points $X', Y'$ exist whose chord $X'Y'$ is in $\Omega$ and thus avoids $K$. One approach might be to construct $H$ as a countable dense set where each pair's connecting chord belongs to $\Omega$. Achieving density alone is insufficient; all chords between any two points in $H$ must be in $\Omega$. Directly constructing such an $H$ presents significant difficulties.
```
</Path>
<Conclusion>
```
The Baire Category Theorem approach for constructing $H$ via intersections (Path 1) lacks a clear method. A step-by-step construction (Path 2) highlights the requirements of density and $K$-avoidance for all chords within $H$, but faces challenges in maintaining the chord property universally during the construction process.
```
</Conclusion>
</Parallel>
```

## D.2 Generated Examples from Multiverse-32B

---

**Multiverse Generation A: Collective Case**

```
<Parallel>
<Goal>
<Outline>
1: Compute Case 1 count.
</Outline>
<Outline>
2: Compute Case 2 count.
</Outline>
</Goal>
<Path>
```
1: In Case 1 the path pattern is H–V–H–V–H, giving three horizontal and two vertical segments. We distribute 8 units into three positive horizontal segments:

$$\binom{8-1}{3-1} = \binom{7}{2} = 21.$$

We distribute 8 units into two positive vertical segments:

$$\binom{8-1}{2-1} = \binom{7}{1} = 7.$$

Total for Case 1:

$$21 \times 7 = 147.$$

```
</Path>
<Path>
```
2: In Case 2 the pattern is V–H–V–H–V, giving three vertical and two horizontal segments. Distribute 8 units into three positive vertical segments:

$$\binom{7}{2} = 21.$$

Distribute 8 units into two positive horizontal segments:

$$\binom{7}{1} = 7.$$

Total for Case 2:

$$21 \times 7 = 147.$$

```
</Path>
<Conclusion>
```
Case 1 yields 147 paths; Case 2 yields 147 paths; overall $147 + 147 = 294$.
```
</Conclusion>
</Parallel>
```

---

**Multiverse Generation B: Selective Case**

```
<Parallel>
<Goal>
<Outline>
1: Count horizontal vs. vertical segments.
</Outline>
<Outline>
2: List possible H/V patterns.
</Outline>
</Goal>
<Path>
```

1: A path from $(0,0)$ to $(8,8)$ uses exactly eight right moves $(R)$ and eight up moves $(U)$. With five segments alternating direction, starting with $R$ forces the pattern

$$R \; - \; U \; - \; R \; - \; U \; - \; R,$$

giving three horizontal segments and two vertical segments. Conversely, starting with $U$ yields

$$U \; - \; R \; - \; U \; - \; R \; - \; U,$$

for three vertical and two horizontal segments. In either case there are four direction changes (one at each segment boundary), and the segment counts differ by one because there are an odd number of segments.
</Path>
<Path>
2: The total move count is 16 ($8R + 8U$). If $H = 3$ and $V = 2$, then the three horizontal segment lengths $h_1 + h_2 + h_3 = 8$ with each $h_i \geq 1$, which has

$$\binom{8-1}{3-1} = \binom{7}{2} = 21$$

compositions. The two vertical segments $v_1 + v_2 = 8$ with $v_j \geq 1$ has

$$\binom{8-1}{2-1} = \binom{7}{1} = 7$$

compositions, for $21 \times 7 = 147$ ways when starting horizontal. Similarly, if $V = 3$ and $H = 2$, there are 21 ways to split 8 into three vertical segments and 7 ways for two horizontal segments, again 147 ways. In total there are $147 + 147 = 294$ segment-length configurations consistent with five alternating segments summing to $(8,8)$.
</Path>
<Conclusion>
A five-segment path must alternate directions, yielding either $(H, V) = (3, 2)$ or $(2, 3)$ segments. There are 147 ways to allocate segment lengths in each case, for 294 total valid patterns.
</Conclusion>
</Parallel>

