# OpenReview forum: "Multiverse: Your Language Models Secretly Decide How to Parallelize and Merge Generation"
_NeurIPS.cc/2025/Conference — NeurIPS 2025 spotlight_

### Official Review · Reviewer_y3b7 · 2025-07-01

**Clarity:** 2
**Significance:** 3
**Originality:** 3
**Rating:** 5
**Confidence:** 3

**Summary:**

The authors present a prompting, fine-tuning, and execution method that enables decomposing problems into parallel components. The system first decomposes the problem into subproblems, then generates them independently, and merges them into a solution for the problem. To fine-tune models, they propose the Multiverse 1K dataset, which contains explicitly annotated parallel branches.

**Questions:**

- Section 5.1 describes the process of creating the dataset, but this is in contrast with Sec 3.2, where the authors claim that the LLMs cannot decompose the problem into parallel branches. What is the key difference between Sec 3.2 and 5.1 that enables the LLMs to generate them? Isn't Sec. 3.2 Just a prompting issue then?
- What are the baselines reported in Tab 2. exactly? Is Autoregressive 32B the same as Qwen2.5-32B-Instruct just fine-tuned on the task in the standard way? Is the difference between Multiverse-32B-zero and Multiverse-32B only the system prompt, or are they also fine-tuned differently? The text says "variant prompted without the 'think in parallel' instruction", but is this done only in test time or also during tuning? Also, if prompted without "think in parallel"  but fine-tuned with "think in parallel" in place and by generating parallel branches, does it still generate parallel branches?
- What is the parallelism metric reported in Tab 2? The caption says "ratio between the total number of generated tokens and the actual generation length", but I don't understand what the difference is between the total number of generated tokens and the generation length. Is it something similar to what I described in the weaknesses?
- How do the authors measure the implicit parallelism for Table 1 and Figure 3? Do they check the examples by hand? Is there some more advanced promoting technique that is used?
- Fig 8: What is P?
- Line 83: what are the varying batch sizes?
- Line 132: For the hidden-state probe, the authors should probably use some internal layer of the model rather than the last.
- Line 154: Why are there square brackets in $x_{[1:3,s]}$ and not in $x_{1,1:6}$?
- Line 204: How does this rewrite work? Is it by prompting or by hand?

**Ethical Concerns:**

["NO or VERY MINOR ethics concerns only"]

**Final Justification:**

The authors cleared the doubts that I had about the paper. I think it is an interesting idea that's worth publishing.

**Limitations:**

The authors discuss the limitation of their work in a satisfactory way.

**Paper Formatting Concerns:**

I have no formatting concerns.

**Quality:**

3

**Strengths And Weaknesses:**

# Strengths

- Interesting and novel idea
- Relatively simple

# Weaknesses
- Some details are not clear. See questions below.
- For most of the examples, the amount of seedup in Table 2 is relatively low. This is surprising given the ~7 average branches reported in Tab 1. Perhaps the authors should consider reporting a parallelism metric where they compute the "parallelness" of the sequence, where they sum the length of the sequential steps, but they use the max of the branches as the length of a parallel block, and they divide this number by the original length.

---

> ### Author Rebuttal · Authors · 2025-07-31
>
> Thank you for reviewing our paper and for your valuable feedback. Below, we address your concerns point by point, and we will revise our paper according to your suggestions. We would appreciate it if you could let us know whether your concerns are addressed by our response. We hope that you might consider raising your score in light of these clarifications.
>
> ---
> **Q1**: How to define the parallelism? Could you provide some parallelism metric? The speedup is unmatched with the number of parallel branches.
>
> **A1**: In this paper, we define the *degree of parallelism* as
>
> $\text{parallelism} = \frac{\text{total tokens generated}}{\text{total tokens generated **sequentailly**}}$.
>
> Because Multiverse can generate multiple tokens per step, its denominator, “sequential tokens”, can be smaller than the total number of generated tokens. This value remains tied to the maximum generation length, which in turn is defined by the highest position id in the generation.
>
> Crucially, the number of parallel branches alone only illustrates the frequency of implicit parallel generation, but does not determine this ratio. What matters is each branch’s *coverage*—the portion of the overall task (generated tokens) that it handles in parallel. In practice, major parallel structures arise within specific subtasks rather than spanning the entire reasoning trace.
>
> To further understand this question, we also compute the parallelism in our Multiverse-1K , which is **16.7 %** on average. This value matches the parallelism and speedup seen at inference time. We will further include this value and clarify their difference in our later version for better understanding.
>
> ---
> **Q2**: Whether LLM can generate in parallel or not? The key difference between section 3.2 and section 5.1
>
> **A2**: The key difference lies in whether the model is creating a parallel structure from scratch or identifying parallelism within an existing structure.
> - In section 3.2, we verified that LLM struggles to generate a parallel thinking trace directly from the instruction prompt. We call this explicit parallelism.
> - In section 5.1, when provided with a pre-existing, sequentially-generated thinking trace, LLM can effectively identify the inherent parallel logic within that trace. We call this implicit parallelism.
>
> ---
> **Q3**: The difference between Multiverse 32B-zero vs Multiverse 32B
>
> **A3**: Thank you for the question. Both Multiverse-32B-zero and Multiverse-32B use the same fine-tuned checkpoint; the only distinction is the inference prompt:
> - Multiverse 32B-zero is prompted with "Let's think step by step."
> - Multiverse 32B is prompted with "Let's think step by step and in parallel."
>
> We included both variants to understand the prompt’s impact on parallelism and overall performance. As shown in Table 2, adding the phrase “and in parallel” preserves accuracy while markedly increasing the degree of parallelism. Therefore, we view this technique as an effective way of modifying model behavior. Moreover, this prompt design also mirrors our training pipeline: autoregressive data were tagged with “Let’s think step by step,” while Multiverse data were tagged with “Let’s think step by step and in parallel.” The correspondence between these training prompts and the observed inference patterns underscores the critical role of prompt engineering in shaping model behaviour with prompt fine-tuning.
>
> ---
> **Q4**: What is the baseline in table 2 exactly? How was the autoregressive model trained?
>
> **A4**: To train the autoregressive baseline, we keep the data mixture, learning rate, number of epochs, and all other hyperparameters identical to those used for Multiverse-32B. The only changes are:
> - Attention mechanism: Multiverse attention is replaced with standard causal attention.
> - Corpus processing: All special tokens are stripped out, and we delete the content in Map phases (<Goal> … </Goal>) and Reduce phases (<Conclusion> … </Conclusion>) so that the remaining text reflects a purely sequential chain-of-thought (CoT) process.
> This is the **main baseline** we hope to compare with in Tab 2.
>
> ---
> **Q5**: How to measure the implicit parallelism in Table 1 and Figure 3? Is it by hand or prompt?
>
> **A5**: Thank you for the question. We measure implicit parallelism with a two-step process that combines LLM automation and human verification:
> -  First, we prompt Gemini-2.5-Pro to identify and list every instance of implicit parallelism.
> -  Next, a human annotator verifies each instance and categorizes it as either a selective or collective branch.
>
> ---
> **Q6**: What is P in figure 8?
>
> **A6**: The variable P represents the degree of parallelism, with the parallel metric clarified in A1. For readability in the current version, we scale it by a factor of 10; thus, P = 11 corresponds to an actual parallelism of 1.1. In the camera-ready version, we will remove this scaling factor, allowing the degree of parallelism to be visualized directly.
>
> ---
> **Q7**: Line 83: What is the varying batch size?
>
> **A7**: The varying batch size corresponds to the experiment detailed in Figure 8b where we tested the speedup under different batch size. The results show that the speedup from multiverse generation is scalable with batch size increase and maintained stable under different settings.
>
> ---
> **Q8**: Line 132: For the hidden states probing experiment, use the internal layer instead of the last layer.
>
> **A8**: Here we provide the probing test of different models in the internal layer.
> Table R1: Probing Test for QWQ-32B
> | Layer | 8 | 16 | 24 | 32 | 40 | 48 | 56 | 64 |
> | :--- | :---: | :---: | :---: | :---: | :---: | :---: | :---: | :---: |
> | Accuracy | 48% | 49% | 52% | 52% | 55% | 48% | 47% | 48% |
>
> Table R2: Probing Test for R1-Distilled 32B
> | Layer | 8 | 16 | 24 | 32 | 40 | 48 | 56 | 64 |
> | :--- | :---: | :---: | :---: | :---: | :---: | :---: | :---: | :---: |
> | Accuracy | 53% | 48% | 54% | 52% | 49% | 55% | 51% | 57% |
>
> Table R3: Probing Test for R1-Distilled 32B
> | Layer | 10 | 20 | 30 | 40 | 50 | 60 | 70 | 80 |
> | :--- | :---: | :---: | :---: | :---: | :---: | :---: | :---: | :---: |
> | Accuracy | 51% | 52% | 49% | 50% | 47% | 46% | 48% | 53% |
>
> The result verifies the conclusion that LLM does not truly understand parallelism.
>
> ---
> **Q9**: Line 154: Why are there square brackets in x[1:3,s] not in x[1,1:6]
>
> **A9**: Thank you for your question! Here, we use x[1:3, s] to combine the variables of $x_1$, $x_2$ and $x_s$. Therefore, the square brackets denotes a list with two separate part. For $x\_{1,1:6}$, we do not use square brackets as we view 1:6 as a whole. We will further add square brackets.
>
> ---
> **Q10**: Line 204: How does the rewrite work? Is it implemented by hand?
>
> **A10**: The entire data curation workflow, including all rewriting steps, was conducted via API calls to the Gemini 2.5 pro model. Here is the detailed prompt for rewrite:
> ```
> # Rewriting Paths in the Structured Reasoning Trajectory
>
> You are given a full structured reasoning trajectory inside a `<Parallel>` block, consisting of:
>
> * one `<Goal>` block with multiple `<Outline>` elements
> * multiple `<Path>` blocks
> * one `<Conclusion>` block.
>
> Some `<Path>` blocks may contain an entire nested `<Parallel>` structure (from `<Parallel>` to `</Parallel>`). These nested blocks should be rewritten using the same rules recursively.
>
> ***
>
> ## For `<Goal>`
>
> * Rewrite each `<Outline>` into a **concise statement of what is being calculated or determined**.
> * Remove any content describing **how** the problem is solved or intermediate reasoning steps.
>
> ***
>
> ## For each `<Path>`
>
> * Keep the original numbering prefix (e.g., `1:`, `2:`).
> * Rewrite the content as a **complete, fluent, and logically self-contained paragraph**.
> * Do **not** use transitional phrases like “First," “Then," “Next," “On the other hand," etc.
> * If the `<Path>` contains **five or fewer sentences**, rewrite them together as a single coherent paragraph, ensuring logical flow and fluency without using transitional phrases.
> * If the `<Path>` contains **more than five sentences**: Rewrite the first five sentences together as a single unit, forming a fluent paragraph. For the remaining sentences, rewrite each one individually, based on its meaning, as clear and fluent standalone statements.
> * If the `<Path>` contains a **nested `<Parallel>` block**, apply all these rules recursively to the nested block.
>
> Each `<Path>` must make sense independently, even if it contains a nested reasoning chain.
>
> ***
>
> ## For `<Conclusion>`
>
> * Rewrite the conclusion as the **most concise and synthesized summary** of the main outcomes from all `<Path>` blocks.
> * You may combine or compare results from different paths, but keep it succinct and direct.
>
> ***
>
> ## Nested `<Parallel>`
>
> * A nested `<Parallel>` may appear only **as a full block inside a `<Path>`**.
> * If a `<Path>` contains a nested `<Parallel>...</Parallel>` block, process that inner block exactly as you would the top-level one:
>     * Rewrite the inner `<Goal>`, `<Path>`, and `<Conclusion>` elements accordingly.
>     * Maintain the XML structure — do not reindent or alter the tag hierarchy.
>
> ```

---

> > ### Comment · Reviewer_y3b7 · 2025-08-02
> >
> > We would like to thank the authors for their response, so I am increasing my score. This cleared most of the questions that I had. Please include the explanations for clarity in the final version of the paper.

---

> > > ### Author Response · Authors · 2025-08-05
> > >
> > > Dear Reviewer y3b7,
> > >
> > > Thank you for your thoughtful re-evaluation and for your trust in the potential impact of our work. We truly appreciate all your suggestions and questions.
> > >
> > > We will carefully incorporate your suggestions and the newly provided information into the final version to further improve clarity and readability.

---

### Official Review · Reviewer_AVNf · 2025-07-02

**Clarity:** 3
**Significance:** 4
**Originality:** 4
**Rating:** 5
**Confidence:** 5

**Summary:**

This paper converts existing single threaded dataset into multi-threaded ones. Furthermore, when fine-tuned on this dataset, the model learns to replicate the behavior on unseen problems, thus realizing the proposed benefits.

**Questions:**

Multiverse attention. I can understand that the positional index being set to the same over the different <path> during training. There is a supposed benefit that the <path> follows directly after the prefix. Please explain why during inference different various paths can share the same positional index for the "reduce".

The argument given that "current LLM cannot parallelize" is rather weak. I do not object to that because whether or not the claim is valid, it does not take away the main contribution of the paper. I do recoomend to either improve the argument or to remove.

I suggest that authors can cite works (don't know if published or not) which aim to allow multiple "paths" to also see each others progress, as opposed to just the common prefix, in the Related Works.

**Ethical Concerns:**

["NO or VERY MINOR ethics concerns only"]

**Final Justification:**

I apologize that I didn't spot this "Final Justification" field when the reviewer panel asked me to revise the review. Took me several attempts to try to find where I needed to revise but couldn't find it.

After the rebuttal I think all the minor concerns - all concerns were minor to begin with - are addressed. I maintain my score of Accept.

**Quality:**

3

**Strengths And Weaknesses:**

Strength:

This is a burgeoning topic that see more and more attempts at addressing. This publication takes a very straightforward approach to get quite satisfactory results. This is the case where simple is good. The S1 dataset should be a boon for people to comprehend, extend, and improve upon this Multiverse.

Though running the inference in a multi-threaded manner, the authors generated the multiverse training data in a single-threaded manner. Although it should be understood by experts in this field, most readers, especially junior ML practitioners, would not immediately comprehend how practical this is.

Weakness:

the "map reduce" analogy is a weak one, I don't think it's that necessary. After all, most of the readers are not and will not be employed in cloud companies, and they don't and shouldn't be expected to - even if they hold a CS degree - instantly understand what map-reduce is. It's very simply a spawn-collect (scatter-gather, or even simpler, fork-join) pattern of programming.

The concept of "plan - spawn - collect" is fairly straightforward. There are certainly many other forms of parallelism.

here are still many unresolved issues in multi-threaded LLM as alluded to in this paper. As this is the first step to multi-threaded LLM, I probably should would refrain from calling this "weakness". I expect any void to be quickly filled.

---

> ### Author Rebuttal · Authors · 2025-07-31
>
> Thank you for your suggestions. We have responded to your questions below and would appreciate it if you could let us know if our response addresses your concerns. We hope that you might consider raising your score in light of these clarifications.
>
> ---
> **Q1**: Instead of “Map-Reduce” why not use spawn-collect (scatter-gather, or even simpler, fork-join)?
>
> **A1**:  Thank you for this suggestion. After revisiting the core definitions of MapReduce and Fork-Join, we now recognize that Fork-Join more accurately captures the type of parallelism enabled by Multiverse on modern hardware. Unlike MapReduce's batch-oriented, fault-tolerant processing across large clusters, Fork-Join's work-stealing scheduler and fine-grained task decomposition better exemplify the low-overhead, multi-core parallel computation with shared KV cache that we leverage. Framing Multiverse within a Fork-Join paradigm is indeed more precise and should facilitate better understanding, especially for a broader audience. We will incorporate this revised terminology in our camera-ready version if the paper is accepted.
>
> ---
> **Q2**:  Explain during inference, why different paths can share the same position ids when “reduce”.
>
> **A2**:  First, although we refer to “shared position IDs,” we do not actually have any computation between different query and key tokens for the same position. This is because softmax attention only considers the distance (i.e., $QK^T$) between token pairs that are not masked, as defined in Equation 1. Thus, the only constraint is that the query token’s positional id must be greater than or equal to the key token’s positional id to ensure the causal relation. By assigning the “Conclusion” token a positional id equal to one more than the maximum id used by any path, we satisfy this causal requirement. (For a deeper discussion of why this works and KV states from different paths can be merged, please see APE [1].)
>
> Second, we employ the same position id assignment rule during both training and inference, ensuring full consistency. This symmetry means the optimization process directly reflects inference behavior, because the model is trained to attend to tokens that share identical positional ids.
>
> Finally, position sharing itself is not new; it has been widely adopted in prior work on LLM length extrapolation [2, 3]. Because a token’s representation combines its hidden state and its positional embedding, modest adjustments to positional ids (even only during inference) have little adverse effect on model performance, as evident by these work.
>
> ---
> **Q3**: The argument that “current LLM can not parallelize” is rather weak
>
> **A3**: We thank the reviewer for highlighting this point and agree that the statement “current LLMs cannot parallelize” is ambiguous. Modern LLMs do exploit parallelism in several aspects, such as prefilling input tokens or serving multiple requests concurrently. Our argument, however, centers on a more specific and fundamental bottleneck: the autoregressive decoding process. This focus is critical because autoregressive modeling remains the dominant architecture underlying today’s leading LLMs.
>
> During generation for a single response, each token is conditioned on all preceding tokens—a dependency represented by $P(\text{token}_n \mid \text{token}_1, \ldots, \text{token}\_{n-1})$. This strict causal dependency prevents parallelization within a single decoding sequence, leading to inherently sequential generation. As a result, latency remains high, particularly in long-form generation tasks such as Chain-of-Thought (CoT) reasoning. Drawing on the concept of goodput [4], we emphasize that overcoming this fundamental bottleneck is critical for improving efficiency in latency-sensitive applications.
>
> In the camera-ready version, we plan to include these details in our introduction for improved clarity. We will also expand the related work section with additional hardware details. Specifically, we will highlight that modern GPUs achieve speedups primarily through parallel computation, yet autoregressive decoding is largely I/O-bound, meaning that increased compute resources do not proportionally reduce latency. This reinforces the importance of rethinking decoding architectures to unlock genuine gains in responsiveness.
>
> ---
> **Q4**: Suggestions about citation on works that aim to allow different paths to see each other during inference.
>
> **A4**: Thank you for your suggestions. We will definitely include work in this direction in our next version, including [5] and [6]. Additionally, we have conducted an ablation study on Multiverse’s sharing mechanism. In this setting, each parallel path can read the tokens produced by the other paths after a fixed interval of k sequential tokens, both during training and inference. As shown in Table R1, this cross-path visibility yields no performance improvement; in fact, overly frequent sharing slightly degrades accuracy, likely because the model becomes confused by mixing KV-cache states from different paths during the generation of each path. (Here, k=$\infty$ means the strategy used in our paper.)
>
> Table R1: Ablation Study between sharing frequency and downstream performance of Multiverse-32B.
> | Task | AIME24 | AIME25 | MATH500 | GPQA-Diamond |
> | :--- | :---: | :---: | :---: | :---: |
> | k=1 |  48.8 | 40.8 | 91.0 | 58.1 |
> | k=8 |  50.4 | 41.7 | 91.8 | 60.6 |
> | k=64 |  51.3 | 43.3 | 92.4 | 60.6 |
> | k=$\infty$ |  51.7 | 42.5 | 92.4 | 61.7 |
>
> ---
> **Reference:**
> [1]. Yang, Xinyu, Tianqi Chen, and Beidi Chen. "Ape: Faster and longer context-augmented generation via adaptive parallel encoding." arXiv preprint arXiv:2502.05431 (2025).
>
> [2]. Jin, Hongye, et al. "Llm maybe longlm: Self-extend llm context window without tuning." arXiv preprint arXiv:2401.01325 (2024).
>
> [3]. Rerope: https://github.com/bojone/rerope
>
> [4]. Zhong, Yinmin, et al. "{DistServe}: Disaggregating prefill and decoding for goodput-optimized large language model serving." 18th USENIX Symposium on Operating Systems Design and Implementation (OSDI 24). 2024.
>
> [5]. Rodionov, Gleb, et al. "Hogwild! inference: Parallel llm generation via concurrent attention." arXiv preprint arXiv:2504.06261 (2025).
>
> [6]. Hsu, Chan-Jan, et al. "Group Think: Multiple Concurrent Reasoning Agents Collaborating at Token Level Granularity." arXiv preprint arXiv:2505.11107 (2025).

---

> > ### Comment · Reviewer_AVNf · 2025-08-07
> >
> > Q1: glad we share views on terminologies
> > Q2: your text was clear, I just didn't read careful enough
> > Q3: I agree
> > Q4: I agree

---

> > > ### Author Response · Authors · 2025-08-08
> > >
> > > We sincerely appreciate the reviewer’s thoughtful feedback on our rebuttal, which provides valuable guidance for further improving our work. We will revise the manuscript to improve the writing by providing clearer definitions and explanations.

---

> ### Comment · Area_Chair_NsRC · 2025-08-03
>
> Dear Reviewer,
>
> Could you please check if the authors’ rebuttal adequately addresses your concerns? If so, kindly acknowledge the rebuttal and provide any additional comments. If not, it would be greatly appreciated it if you could engage in a discussion with the authors. Your input at this stage is essential to the review process. Thank you very much for your time and effort!
>
> AC

---

> ### Author Response · Authors · 2025-08-05
>
> Dear Reviewer,
>
> I hope this message finds you well. As the discussion period is nearing its end, I wanted to ensure we have addressed all your concerns satisfactorily. If there are any additional points or feedback you'd like us to consider, please let us know. Your insights are invaluable to us, and we're eager to address any remaining issues to improve our work.
>
> Thank you for your time and effort in reviewing our paper.

---

### Official Review · Reviewer_RkKR · 2025-07-03

**Clarity:** 3
**Significance:** 3
**Originality:** 4
**Rating:** 5
**Confidence:** 4

**Summary:**

This paper introduces Multiverse, a test-time scaling framework for LLM that enables parallel generation. Motivated by the observation that autoregressive models often generate reasoning trajectories with latent parallelizable branches, the authors propose a MapReduce framework that decomposes generation into three stages: Map, Process, and Reduce. The authors design three components: data curation, architecture design, and system implementation. Empirically. the authors demonstrate improvements on mathematical reasoning tasks.

**Questions:**

1. How important is each of the three components—data, Multiverse attention, scheduler? For example, would just fine-tuning Qwen2.5 on the structured outputs suffice?

2. In Line 126, how does the prompting test work? Can the authors clarify this?

In general, I lean toward acceptance despite the mentioned weaknesses and questions.

**Ethical Concerns:**

["NO or VERY MINOR ethics concerns only"]

**Final Justification:**

I appreciate the authors' effort in showing every detailed response. My concerns are largely addressed. Overall, it is a strong paper, and I maintain my recommendation of a clear accept.

**Limitations:**

Yes.

**Paper Formatting Concerns:**

No major formatting issues.

**Quality:**

3

**Strengths And Weaknesses:**

## Strengths
1. The paper demonstrates the inherent parallelizable features of autoregressive generation. The proposed method is novel; the MapReduce framework is elegant and practically useful.
2. The paper is well written with rich illustrations and clean empirical and structural analysis.


## Weaknesses
1. The empirical improvement of the proposed method appears limited (Table 2, Figure 7). Error bars are not provided, so it's unclear whether the gains are statistically significant or within expected variance.
2. Ablations are missing: How important is each of the three components—data, Multiverse attention, scheduler? For example, would just fine-tuning Qwen2.5 on the structured outputs suffice?

---

> ### Author Rebuttal · Authors · 2025-07-31
>
> Thank you for your suggestions. We have responded to your questions below and would appreciate it if you could let us know if our response addresses your concerns. We hope that you might consider raising your score in light of these clarifications.
>
> ---
> **Q1**: The empirical improvement of the proposed method appears limited.
>
> **A1**: We appreciate your question and agree that the current empirical improvement is limited. This outcome is directly linked to the degree of parallelism in our training data, Multiverse-1K. Our statistics shows that Multiverse-1K has an average parallelism degree of 1.16 which matches the parallelism observed at inference. This design choice aligns with our core research objective: rather than artificially increasing parallelism, we focus on externalizing the implicit parallel generation capabilities in modern autoregressive LLMs. Therefore, the observed parallelism degree of 1.16 represents the natural baseline of parallel reasoning these models inherently possess, which our Multiverse seeks to make explicit and accessible.
>
> ---
> **Q2**: Error bars are not provided thus it is unclear the gains are statistically significant.
>
> **A2**: Thank you for your suggestions. Here we provide our results with error bars for Tab 2, Figure 7 and Figure 8b.
>
> Table 1: Reasoning Performance
> | **Model / Metric** | **AIME24** pass@1 | **AIME24** #parallel | **AIME25** pass@1 | **AIME25** #parallel | **MATH500** pass@1 | **MATH500** #parallel | **GPQA-Diamond** pass@1 | **GPQA-Diamond** #parallel |
> | :--- | :---: | :---: | :---: | :---: | :---: | :---: | :---: | :---: |
> | s1-32B | $35.4 \pm 1.7$ | 1.00 | $25.8 \pm 1.5$ | 1.00 | $88.6 \pm 0.5$ | 1.00 | $48.0 \pm 0.4$ | 1.00 |
> | s1.1-32B | $52.9 \pm 1.7$ | 1.00 | $41.7 \pm 1.7$ | 1.00 | $93.4 \pm 0.4$ | 1.00 | $60.3 \pm 0.6$ | 1.00 |
> | Qwen2.5-32B-Instruct | $15.8 \pm 1.5$ | 1.00 | $10.4 \pm 1.2$ | 1.00 | $80.4 \pm 0.2$ | 1.00 | $47.0 \pm 0.4$ | 1.00 |
> | Autoregressive-32B | $51.3 \pm 1.9$ | 1.00 | $42.9 \pm 1.6$ | 1.00 | $92.8 \pm 0.4$ | 1.00 | $61.6 \pm 0.7$ | 1.00 |
> | Multiverse-32B-zero | $52.1 \pm 1.7$ | $1.07 \pm 0.02$ | $44.2 \pm 1.8$ | $1.05 \pm 0.01$ | $91.8 \pm 0.2$ | $1.05 \pm 0.01$ | $62.1 \pm 0.5$ | $1.06 \pm 0.01$ |
> | Multiverse-32B | $52.9 \pm 2.1$ | $1.24 \pm 0.03$ | $44.2 \pm 1.5$ | $1.18 \pm 0.03$ | $92.4 \pm 0.4$ | $1.15 \pm 0.01$ | $61.7 \pm 0.5$ | $1.17 \pm 0.02$ |
>
> Table 2: GPQA-Diamond Performance with control budget
> | Model | 1k | 1.5k | 2k | 2.5k | 3k | 3.5k | 4k | 4.5k |
> | :--- | :---: | :---: | :---: | :---: | :---: | :---: | :---: | :---: |
> | Multiverse | $32.8 \pm 0.4$ | $35.9 \pm 0.5$ | $40.9 \pm 0.5$ | $45.5 \pm 0.4$ | $48.5 \pm 0.6$ | $50.5 \pm 0.5$ |$52.0 \pm 0.4$ | $53.6 \pm 0.5$ |
> | Autoregressive | $29.3 \pm 0.4$ | $33.84 \pm 0.3$ | $37.88 \pm 0.4$ | $41.27 \pm 0.4$ | $45.96 \pm 0.3$| $49.2 \pm 0.6$ | $51.0 \pm 0.6$ | $54.2 \pm 0.7$ |
>
> Table 3: Math 500 Performance with control budget
> | Model | 1k | 1.5k | 2k | 2.5k | 3k | 3.5k | 4k |
>  | :--- | :---: | :---: | :---: | :---: | :---: | :---: | :---: |
>  | Multiverse | $37.0 \pm 0.3$ | $54.4 \pm 0.2$ | $63.6 \pm 0.4$ | $70.5 \pm 0.2$ | $76.5 \pm 0.3$ | $81.8 \pm 0.2$ | $85.1 \pm 0.3$ |
>  | Autoregressive | $35.2 \pm 0.2$ | $52.8 \pm 0.4 $ | $62.0 \pm 0.3$ | $69.6 \pm 0.4$ | $74.8 \pm 0.4$ | $79.6 \pm 0.3$ | $84.3 \pm 0.4$ |
>
> Table 4: Full Speedup Results Across Varying Degrees of Parallelism and Batch Sizes
> | #Parallel | bsz=4 | bsz=8 | bsz=16 | bsz=32 | bsz=64 | bsz=128 |
> | :--- | :---: | :---: | :---: | :---: | :---: | :---: |
> | 1.1 | $1.07 \pm 0.02$ | $1.13 \pm 0.02$ | $1.06 \pm 0.01$ | $1.07 \pm 0.01$ | $1.08 \pm 0.02$ | $1.15 \pm 0.02$ |
> | 1.2 | $1.20 \pm 0.02$ | $1.18 \pm 0.02$ | $1.18 \pm 0.01$ | $1.18 \pm 0.02$ | $1.19 \pm 0.04$ | $1.22 \pm 0.02$ |
> | 1.3 | $1.31 \pm 0.02$ | $1.25 \pm 0.03$ | $1.32 \pm 0.02$ | $1.31 \pm 0.01$ | $1.31 \pm 0.02$ | $1.35 \pm 0.02$ |
> | 1.5 | $1.47 \pm 0.03$ | $1.46 \pm 0.02$ | $1.54 \pm 0.02$ | $1.51 \pm 0.02$ | $1.50 \pm 0.01$ | $1.55 \pm 0.03$ |
>
> ---
> **Q3**: Ablation study is missing. How important is each of the three components?
>
> **A3**: In our multiverse framework, the three core components—Data, Algorithm, and Engine—are intricately combined. A critical issue arises when fine-tuning the model directly on structured outputs: this process creates a mismatch between the position IDs during fine-tuning  and those during inference stage, leading to significant performance degradation.
>
> ---
> **Q4**: Ablation study is missing. How important is each of the three components?
>
> **A4**: Here we provide the full prompt we use during the prompt test.
> ```
> **Core Directive**
> You are an expert mathematician and reasoner. Your primary mode of operation is to solve problems efficiently.
> When you think, you can process information in parallel to solve multiple parts of a problem at once. This requires you to first analyze the structure of any given problem.
> **Conditional Logic Mandate:**
> During your Chain of Thought thinking progress, you should think in parallel. Specifically determine if the problem can be decomposed into independent, non-sequential sub-tasks.
> * **If the problem is decomposable**, you should use the Parallel Processing Format defined below.
> * **If the problem is inherently sequential** or too simple for decomposition, solve it using a standard, linear step-by-step explanation.
> You are to make this decision yourself. Do not ask for permission.
> ---
> **Parallel Processing Format**
> When a problem is decomposable, you must structure your entire thought process as follows:
> ```xml
> <Parallel>
>   <Goal 1>
>     Define the first independent objective.
>   </Goal>
>   <Goal 2>
>     Define the second independent objective.
>   </Goal>
>
>   <Path 1>
>     Execute the steps to achieve Goal 1. Show your work.
>   </Path>
>   <Path 2>
>     Execute the steps to achieve Goal 2. Show your work.
>   </Path>
>
>   <Conclusion>
>     Synthesize the results from all Paths to generate the final answer.
>   </Conclusion>
> </Parallel>
> ```
> We observed the reasoning generation with this prompt and analyzed the parallelism within the chain of thought.

---

> > ### Comment · Reviewer_RkKR · 2025-08-01
> >
> > Thank you for the detailed response. I maintain my score of accept.

---

> > > ### Author Response · Authors · 2025-08-08
> > >
> > > Thank you for your kind response and for taking the time to review our work. We sincerely appreciate your constructive comments, which have greatly helped us improve the paper. We’re glad to hear that the rebuttal addressed your concerns, and we appreciate your support.

---

### Official Review · Reviewer_eCfu · 2025-07-05

**Clarity:** 3
**Significance:** 2
**Originality:** 3
**Rating:** 3
**Confidence:** 4

**Summary:**

This paper proposes Multiverse, a natively parallel generative model based on the MapReduce paradigm. Autoregressive large language models (AR-LLMs) are constrained by sequential generation, while non-AR models like diffusion models struggle with complex tasks. Multiverse features three stages: Map for adaptive task decomposition, Process for parallel subtask execution, and Reduce for lossless result merging.

The model co-designs data, algorithm, and system: Multiverse 1K dataset transforms sequential reasoning into parallel structures via an LLM-assisted pipeline; Multiverse Attention modifies attention masks for parallel generation; Multiverse Engine enables dynamic scheduling between sequential and parallel modes.

**Questions:**

In the experimental section, are there more settings provided, such as the code? Additionally, the scores on AIME seem too low. Is it difficult to interpret the effectiveness of the method?
At a parallelism ratio of 1.3, the model achieves an 18.5% average speedup. the results seems not strong enough.

**Ethical Concerns:**

["NO or VERY MINOR ethics concerns only"]

**Limitations:**

The experiments is not strong enough, with less than 2 times speed up. and the baseline is also not strong,

**Quality:**

3

**Strengths And Weaknesses:**

Unlike text diffusion models that often rely on rigid parallelization, Multiverse empowers language models to autonomously decide parallel generation strategies, leveraging GPU acceleration through dynamic task decomposition. This self-adaptive parallelism not only optimizes computational efficiency but also preserves logical coherence in complex reasoning, marking a significant advancement in bridging sequential AR-LLMs with parallel hardware capabilities.

---

> ### Author Rebuttal · Authors · 2025-07-31
>
> We would like to sincerely thank reviewer eCfu for the insightful and valuable comments! They are invaluable for further improving the quality of our paper. We will revise our manuscript in the next version to address all of your concerns. We hope that you might consider raising your score in light of these clarifications.
>
> ---
> **Q1**: In the experimental section, are there more settings provided, such as the code?
> **A1**: Thank you for your suggestions. We first need to clarify that our experiments in the paper follow the setups outlined in the S1 paper, which cover both training and evaluation of math and scientific reasoning. This setup is standard and ensures coherence between our training and inference. To address your concern, we further conduct experiments on IFEval and LiveCodeBench as instruction-following and code benchmarks. As shown in Table R1, our method can perform on-par with autoregressive models on these benchmarks. However, they do not increase the performance over base model due to the inconsistency of training and test domains.
>
> Table R1: Experimental Results of various models on instruction-
> | Metric | Qwen2.5-32B-Instruct | s1.1-32B | Autoregressive-32B | Multiverse 32B |
> | :--- | :---: | :---: | :---: | :---: |
> | IFEval | 79.1 | 57.4 | 57.1 | 57.9 |
> | LiveCodeBench(v6) | 39.1 | 39.1 | 38.6 | 39.4 |
>
> ---
> **Q2**: Aime score seems low. Is it difficult to interpret the effectiveness of the method?
> **A2**: We emphasize that this work focuses on developing novel model architectures for parallel generation instead of improving reasoning ability, with Multiverse being the only open-source non-autoregressive model to achieve an AIME score above 50—setting a new standard in this area. For a fair comparison with autoregressive models, we adopt the s1 experimental setting, training on the same corpus (with CoT rewriting), and demonstrate performance comparable to s1.1 while achieving significantly higher efficiency. Given that s1.1 was released only two months before our submission and is a strong reasoning model trained purely with supervised fine-tuning, we consider it a reasonable and fair baseline. Although larger datasets, model scales, or compute budgets could further enhance Multiverse, these resources exceed typical academic constraints and lie orthogonal to our architectural design. In this context, we believe that achieving an AIME score above 50 thus represents both a rigorous standard and clear differentiation of model quality.
>
> ---
> **Q3**: At a parallelism ratio of 1.3, the speedup is 18.5% which seems not strong enough.
> **A3**: As discussed in lines 290-291, the 18.5% speedup is an average across data exhibiting parallelism from 1.0 to 1.3 instead of 1.3 only. Therefore, our speedup should have a strong correlation with the parallelism ratio, which can be further verified in our Figure 8(b).

---

> ### Comment · Area_Chair_NsRC · 2025-08-03
>
> Dear Reviewer,
>
> Could you please check if the authors’ rebuttal adequately addresses your concerns? If so, kindly acknowledge the rebuttal and provide any additional comments. If not, it would be greatly appreciated it if you could engage in a discussion with the authors. Your input at this stage is essential to the review process. Thank you very much for your time and effort!
>
> AC

---

### Decision · Program_Chairs · 2025-09-17

**Decision:**

Accept (spotlight)

**Comment:**

This paper introduces a test-time scaling framework for LLMs that enables decomposing problems into parallel components. The authors also propose the Multiverse-1K dataset for training purposes and present empirical improvements on mathematical reasoning tasks.

Most reviewers find this work novel and value it as the first attempt at multi-threaded LLMs. The proposed dataset also seems to have great potential for future research. Some concerns were raised regarding performance improvements, error bars, and other minor details, but these were well addressed by the authors during the rebuttal.

Given the above, the AC recommends acceptance. This recommendation is based on down-weighting the comments from Reviewer eCfu due to their late response to the rebuttal.